# Massively Multilingual Corpus of Sentiment Datasets and Multi-faceted Sentiment Classification Benchmark

**Łukasz Augustyniak**
WUST (Wrocław University of Science and Technology)
lukasz.augustyniak@pwr.edu.pl

**Szymon Woźniak**
Brand24 AI

**Marcin Gruza**
Brand24 AI, WUST

**Piotr Gramacki**
Brand24 AI, WUST

**Krzysztof Rajda**
Brand24 AI, WUST

**Mikołaj Morzy**
Poznań University of Technology

**Tomasz Kajdanowicz**
WUST

## Abstract

Despite impressive advancements in multilingual corpora collection and model training, developing large-scale deployments of multilingual models still presents a significant challenge. This is particularly true for language tasks that are culture-dependent. One such example is the area of multilingual sentiment analysis, where affective markers can be subtle and deeply ensconced in culture. This work presents the most extensive open massively multilingual corpus of datasets for training sentiment models. The corpus consists of 79 manually selected datasets from over 350 datasets reported in the scientific literature based on strict quality criteria. The corpus covers 27 languages representing 6 language families. Datasets can be queried using several linguistic and functional features. In addition, we present a multi-faceted sentiment classification benchmark summarizing hundreds of experiments conducted on different base models, training objectives, dataset collections, and fine-tuning strategies.

## 1 Introduction

Consider a hotel booking service that allows its customers to post reviews. You have found just the perfect accommodation to stay for a couple of days with your family, but you browse through the reviews section of the website to check the experiences of former guests. Suddenly, you encounter a review in Polish: "*hotel jak hotel, mogło być gorzej.*" This review has the following sentiment scores[1]: $s_{neg} = 0.44, s_{neu} = 0.44, s_{pos} = 0.12$. Intrigued by the ambiguity of scores, you translate the review into English: "*hotel like a hotel, all in all, it could have been worse,*" which is scored as $s_{neg} = 0.80, s_{neu} = 0.16, s_{pos} = 0.04$. Apparently, the stereotypically pessimistic Polish outlook on life gets lost in translation. The next review is written in Czech: "*ok, ale nic zajímavého*" with scores $s_{neg} = 0.32, s_{neu} = 0.54, s_{pos} = 0.14$. After translating into English ("*ok, but nothing interesting*") the sentiment classification model scores the review as negative ($s_{neg} = 0.50, s_{neu} = 0.37, s_{pos} = 0.13$). After your stay, you decide to add a very positive review of the hotel ("*it was a killer place to stay*", $s_{neg} = 0.03, s_{neu} = 0.05, s_{pos} = 0.92$). You would be very surprised to learn that the Czechs would be quite confused about your opinion ("*to bylo vražedné*

---

[1]Sentiment scores in this paragraph are produced by the multilingual `cardiffnlp/twitter-xlm-roberta-base-sentiment` model [9]

*místo k pobytu*", $s_{neg} = 0.51, s_{neu} = 0.09, s_{pos} = 0.39$), while the Poles would stay away from the hotel at all costs ("*to było zabójcze miejsce na pobyt*", $s_{neg} = 0.78, s_{neu} = 0.09, s_{pos} = 0.13$).

multilingual services become ubiquitous in the modern global economy. As more websites begin to offer automatic translation of content, users do not bother to express themselves in the *lingua franca* of the Web, writing instead in their native languages. Despite impressive advancements in automatic translation, many NLP tasks are still difficult in the multilingual setting. And sentiment classification is among the most challenging. The expression of sentiment is highly culture-dependent [31]. The emotional valence of individual words, the presence of specific phrasemes, and the expectations around sentiment values make sentiment classification across languages a demanding task.

Models performing sentiment classifications have to cope with two independent sources of variance in the input data: cultural expressions of sentiment and errors in automatic translations. In addition, the productization of sentiment classification leads to several engineering choices which influence the efficiency of the model:

- *single multilingual model vs. dedicated monolingual models*: deploying a single model in a production environment is much easier than orchestrating an ensemble of models,

- *training vs. fine-tuning*: sentiment classification model can be trained from scratch, both in the multi- and monolingual regimes, and the alternative is the fine-tuning of a general-purpose pre-trained language model with sentiment data,

- *transfer learning between domains*: a sentiment classification model trained for a specific domain (e.g., book reviews) can be transferred to another domain (e.g., hotel reviews) under the assumption that sentiment expressions in a given language remain independent of the subject of sentiment,

- *transfer learning between languages*: one may hypothesize that related languages can utilize similar sentiment expression mechanisms regarding grammar, punctuation, and vocabulary. In theory, it is possible to use training data in language $L_1$, fine-tune a multilingual model with this data, and as a result of fine-tuning, improve the performance of the model on language $L_2$, provided languages $L_1$ and $L_2$ are sufficiently similar.

Working with multilingual datasets and models opens another opportunity. Most NLP research focuses on 20 major languages. Many languages native to significant human populations are not adequately studied. Although the term *low-resource language* is not precisely defined and can be understood in terms of computerization, privilege, the abundance of resources, or density [47], the existence of a large chasm between languages with respect to linguistic resources is apparent. Our work aims at supporting low-resource languages in performant sentiment classification. Finally, there is a lively debate in the scientific community about the inherent ability of neural models to handle general linguistic phenomena. This paper tries to answer this question in the domain of sentiment classification.

Contribution presented in this paper is threefold:

- *multilingual corpus*: we present the largest multilingual collection of sentiment classification datasets consisting of 79 high-quality datasets covering 27 languages. The collection of the corpus and detailed descriptions of collected datasets are presented in Section 3.

- *multi-faceted benchmark*: the dataset is supplemented with a benchmark containing detailed run statistics of hundreds of experiments representing different training and testing scenarios. All details relevant to the benchmark are presented in Section 4.

- *library for dataset access*: all datasets in the multilingual corpus are publicly available via a library compatible with the HuggingFace library, along with the ability to filter and select datasets, verify their licenses, etc[2].

## 2 Linguistic typology

The similarity of languages and the main aspects of their differences is the field of study of language typology. The differences between languages can be phonological (differences in sounds used

---

[2]https://huggingface.co/datasets/Brand24/mms

by languages), syntactic (differences in language structures), lexical (differences in vocabulary), and theoretical (differences characterized as general properties of languages). Linguistic typology analyzes the current state of languages and is often contrasted with genealogical linguistics. The latter is concerned with historical relationships between languages established via historical records or with the help of comparative linguistics. The main focus of genealogical linguistics resolves around *language families* and *language genera*. The term *language family* refers to a group of languages sharing pedigree from a common ancestral language (the *proto-language*). As of today, linguists define over 7000 languages categorized into 150 families [15]. The largest families of languages include Indo-European, Sino-Tibetan, Turkic, Afro-Asiatic, Nilo-Saharan, Niger-Congo, and Eskimo-Aleut [24]. Main families are further divided into branches called *genera*. Examples of genera within the Indo-European family of languages include Slavic, Romance, Germanic, and Indic. This division of language families into genera closely follows the genetic family of humankind as attested by DNA similarity [43]. Some language families do not produce distinctive genera but form dialect continua defined by mutual intelligibility. Finally, an important concept is that of a *sprachbund*, a geographical area occupied by languages sharing linguistic features. The linguistic similarities within a sprachbund result from cultural exchange and geographical contact rather than by chance or common origin, as described by Thomason and Kaufman [85].

Languages can be described using hundreds of linguistic features. World Atlas of Language Structures [27] lists almost 200 different features. Since our work focuses on sentiment classification, we select 10 features that seem to be the most relevant to the task of sentiment expression. These features are:

1. *definite article*: the morpheme associated with nouns and used to code the uniqueness or definiteness of a concept; almost half of the languages do not use the definite article.

2. *indefinite article*: the morpheme used together with nouns to signal that the related concept is unknown to the hearer; half of the languages do not use the indefinite article, some languages use a separate article, and some use the numeral "one" as the indefinite article.

3. *number of cases*: morphological cases are a common way to express various relationships between words; human languages vary greatly in the number of cases used by a language.

4. *order of subject, verb, and object*: some languages have the strict ordering of words, while languages that convey semantics through inflection may be much looser with the ordering; half of all languages use the SOV (subject-object-verb) ordering, one third uses the SVO (subject-verb-object) ordering, and a small fraction of languages use the remaining VSO, VOS, OVS, and OSV orderings. Interestingly, around 13% of world languages do not have any fixed word order.

5. *negative morphemes*: negative morpheme is used to signal clausal negation in declarative sentences; this is usually achieved using a negative affix or a negative particle.

6. *polar questions*: a polar question is a question with only yes/no answers; these questions can be built using question particles, interrogative morphology, or intonation only.

7. *position of the negative morpheme*: languages differ by the position of the negative morpheme in relation to subjects and objects, with many variants such as SNegVO, NegSVO, SVNegO, obligatory and optional double negations, etc.

8. *prefixing vs. suffixing*: languages differ significantly in their use of prefixes versus suffixes in inflectional morphology.

9. *coding of nominal plurals*: two major types of plural coding are present in languages, either by changing the morphological form of the noun or by using a plurality indicator morpheme somewhere in the noun phrase.

10. *grammatical genders*: there is significant variability among languages with respect to the number of grammatical genders, some languages do not use the concept at all, and some languages may have 5 or more grammatical genders.

All language features mentioned above are available as filtering features in our library. Thus, when training a sentiment classifier using our dataset, one may download different facets of the collection. For instance, one can download all datasets in Slavic languages in which polar questions are formed using the interrogative word order (Listing 1) or download all datasets from the Afro-Asiatic language family with no morphological case-making (Listing 2).

# 3 Datasets

## 3.1 Quality criteria

The initial pool of sentiment datasets has been gathered using extensive search and consisted of 345 datasets found through Google Scholar, GitHub repositories, and the HuggingFace datasets library. This initial pool of datasets has been manually filtered based on the following set of quality assurance criteria:

1. *strong annotations*: we have rejected datasets containing weak annotations (e.g., datasets with labels based on emoji occurrence or generated automatically through classification by machine learning models) due to an extensive amount of noise [58].

2. *well-defined annotation protocol*: we have rejected datasets without sufficient information about the annotation protocol (e.g., whether annotation was manual or automatic, number of annotators) to avoid merging datasets with contradicting annotation instructions.

3. *numerical ratings*: we have accepted datasets with numerical ratings, mapping Likert-type 5-point scales into three class sentiment labels as follows: ratings 1 and 2 were mapped to *negative*, rating 3 was mapped to *neutral*, and ratings 4 and 5 were mapped to *positive*.

4. *three classes only*: we have rejected datasets annotated with binary sentiment labels as their performance in three class settings was unsatisfactory.

5. *monolingual datasets*: when a dataset contained samples in multiple languages, we opted to divide it into independent datasets in constituent languages.

## 3.2 Pre-processing of datasets

Despite quality assurance criteria described in Section 3.1, the datasets still contained conflicting entries, i.e., duplicated records with different sentiment labels. We have cross-referenced all datasets to identify conflicts and have made the data coherent using majority-label voting. Finally, labeling and rating schemes of all datasets have been mapped to a 3-class scheme with *negative*, *neutral*, and *positive* labels only. For datasets with emotional annotations, we mapped positive emotions (joy, happiness) into positive sentiment and negative emotions (fear, sadness, disgust, anger) into negative sentiment. Texts with ambiguous emotions like anticipation and surprise were discarded. This pre-processing pipeline resulted in 79 datasets containing $6\,164\,762$ text samples. Most of the datasets are in English ($2\,330\,486$ samples across 17 datasets), Arabic ($932\,075$ samples across 9 datasets), and Spanish ($418\,712$ samples across 5 datasets). The datasets represent four different domains: social media (44 datasets), reviews (24 datasets), news (5 datasets), and others (6 datasets). In addition, all datasets were processed by the `cleanlab` library to produce a self-confidence label-quality score for each data point.

Exhaustive testing of all configurations of the benchmark is unfeasible, but the total lack of baseline is unacceptable, either. We have decided to test the benchmark dataset by manually compiling a strong baseline. The main rationale behind this effort was the lack of coherence in annotation guidelines between considered datasets. Our baseline is built using strict annotation guidelines constructed iteratively over annotation batches, resulting in highly coherent annotations. The baseline dataset consists of $3\,911$ short text samples (trimmed to 350 characters) in Polish and English, annotated independently by 3 annotators fluent in these languages. Baseline texts cover multiple domains, such as social media, news sites, blogs, and Internet forums. For each instance, the majority label assigned by the annotators has been stored. The inter-rater agreement is $\kappa = 0.665$ (average Cohen's kappa between three pairs of annotators) and $\alpha = 0.666$ (Krippendorff's alpha).

One may think of the baseline dataset as representative of current sentiment model training, where researchers have to build domain-aligned corpora of various languages and annotate them. The comparison of results between the benchmark and the baseline datasets is a proxy of the performance trade-offs should one decide to use our benchmark dataset to train domain and language-specific sentiment classifiers.

Table 1: Summary of the corpus. Categories: N - news, R - reviews, SM - social media, O - other

| | #datasets | category | | | | #samples | | | mean | |
|---|---|---|---|---|---|---|---|---|---|---|
| | | N | R | SM | O | NEG | NEU | POS | #words | #chars |
| English | 17 | 3 | 4 | 6 | 4 | 304,939 | 290,823 | 1,734,724 | 62 | 339 |
| Arabic | 9 | 0 | 4 | 4 | 1 | 138,899 | 192,774 | 600,402 | 52 | 289 |
| Spanish | 5 | 0 | 3 | 2 | 0 | 108,733 | 122,493 | 187,486 | 26 | 150 |
| Chinese | 2 | 0 | 2 | 0 | 0 | 117,967 | 69,016 | 144,719 | 60 | 80 |
| German | 6 | 0 | 1 | 5 | 0 | 104,667 | 100,071 | 111,149 | 26 | 171 |
| Polish | 4 | 0 | 2 | 2 | 0 | 77,422 | 62,074 | 97,192 | 19 | 123 |
| French | 3 | 0 | 1 | 2 | 0 | 84,187 | 43,245 | 83,199 | 28 | 159 |
| Japanese | 1 | 0 | 1 | 0 | 0 | 83,982 | 41,979 | 83,819 | 61 | 101 |
| Czech | 4 | 0 | 2 | 2 | 0 | 39,674 | 59,200 | 97,413 | 34 | 212 |
| Portuguese | 4 | 0 | 0 | 4 | 0 | 56,827 | 55,165 | 45,842 | 11 | 63 |
| Slovenian | 2 | 1 | 0 | 1 | 0 | 33,694 | 50,553 | 29,296 | 41 | 269 |
| Russian | 2 | 0 | 0 | 2 | 0 | 31,770 | 48,106 | 31,054 | 11 | 70 |
| Croatian | 2 | 1 | 0 | 1 | 0 | 19,757 | 19,470 | 38,367 | 17 | 116 |
| Serbian | 3 | 0 | 2 | 1 | 0 | 25,089 | 32,283 | 18,996 | 44 | 269 |
| Thai | 2 | 0 | 1 | 1 | 0 | 9,326 | 28,616 | 34,377 | 22 | 381 |
| Bulgarian | 1 | 0 | 0 | 1 | 0 | 13,930 | 28,657 | 19,563 | 12 | 86 |
| Hungarian | 1 | 0 | 0 | 1 | 0 | 8,974 | 17,621 | 30,087 | 11 | 83 |
| Slovak | 1 | 0 | 0 | 1 | 0 | 14,431 | 12,842 | 29,350 | 13 | 98 |
| Albanian | 1 | 0 | 0 | 1 | 0 | 6,889 | 14,757 | 22,638 | 13 | 91 |
| Swedish | 1 | 0 | 0 | 1 | 0 | 16,266 | 13,342 | 11,738 | 14 | 94 |
| Bosnian | 1 | 0 | 0 | 1 | 0 | 11,974 | 11,145 | 13,064 | 12 | 76 |
| Urdu | 1 | 0 | 0 | 0 | 1 | 5,239 | 8,585 | 5,836 | 13 | 69 |
| Hindi | 1 | 0 | 0 | 1 | 0 | 4,992 | 6,392 | 5,615 | 26 | 128 |
| Persian | 1 | 0 | 1 | 0 | 0 | 1,602 | 5,091 | 6,832 | 21 | 104 |
| Italian | 2 | 0 | 0 | 2 | 0 | 4,043 | 4,193 | 3,829 | 16 | 103 |
| Hebrew | 1 | 0 | 0 | 1 | 0 | 2,279 | 243 | 6,097 | 22 | 110 |
| Latvian | 1 | 0 | 0 | 1 | 0 | 1,378 | 2,618 | 1,794 | 20 | 138 |

# 4  Multi-faceted benchmark

As we have explained in Section 1, the deployment of a multilingual sentiment classifier can be evaluated using several criteria leading to different architecture choices. Thus, we do not publish a single benchmark, but we aggregate the results along several dimensions. The benchmark available at `https://huggingface.co/spaces/Brand24/mms_benchmark`, allows to compare models according to:

- *number of languages*: multilingual models with monolingual models,

- *training procedure*: models trained from scratch vs. fine-tuning,

- *domain language*: language to which a model is applied,

- *data modality*: news, reviews, social media,

- *knowledge transfer*: transfer between languages, transfer between domains.

Table 2 presents the list of models included in the benchmark. For each model, we include the number of parameters and languages used in pre-training and the base model. The results presented in the benchmark reflect three possible scenarios of model deployment. A pre-trained model can be used to generate text representation only. In this scenario, denoted *head-linear* (HL), a model serves as a feature extractor followed by a small linear classification head. In the second scenario, the linear classification head is replaced by a BiLSTM classifier operating on features extracted by the pre-trained model. We refer to this scenario as *head-bilstm* (HB). Finally, each pre-trained transformer-based model (with the exception of mUSE-transformer) has been fine-tuned to the sentiment classification task. We refer to this scenario as *fine-tuning* (FT).

Table 2: Models included in the benchmark

| Model | #params | #langs | base | reference |
|---|---|---|---|---|
| mT5 | 277M | 101 | T5 | Xue et al. [93] |
| LASER | 52M | 93 | BiLSTM | Artetxe and Schwenk [5] |
| mBERT | 177M | 104 | BERT | Devlin et al. [26] |
| MPNet | 278M | 53 | XLM-R | Reimers and Gurevych [64] |
| XLM-R-dist | 278M | 53 | XLM-R | Reimers and Gurevych [64] |
| XLM-R | 278M | 100 | XLM-R | Conneau et al. [22] |
| LaBSE | 470M | 109 | BERT | Feng et al. [30] |
| DistilmBERT | 134M | 104 | BERT | Sanh et al. [73] |
| mUSE-dist | 134M | 53 | DistilmBERT | Reimers and Gurevych [64] |
| mUSE-transformer | 85M | 16 | transformer | Yang et al. [95] |
| mUSE-cnn | 68M | 16 | CNN | Yang et al. [95] |

Fine-tuning - F1 score per language

| | lang | ds | all | en | ar | es | zh | de | pl | fr | ja | cs | pt | sl | ru | hr | sr | th | bg | hu | sk | sq | sv | bs | ur | hi | fa | it | he | lv |
|---|---|---|---|---|---|---|---|---|---|---|---|---|---|---|---|---|---|---|---|---|---|---|---|---|---|---|---|---|---|---|
| XLM-R | 61 | 61 | 68 | 70 | 66 | 64 | 64 | 72 | 67 | 70 | 68 | 65 | 44 | 58 | 66 | 55 | 50 | 65 | 61 | 62 | 62 | 45 | 61 | 55 | 50 | 51 | 63 | 61 | 58 | 65 |
| LaBSE | 60 | 62 | 67 | 69 | 67 | 63 | 61 | 71 | 67 | 70 | 64 | 63 | 42 | 59 | 66 | 58 | 50 | 63 | 61 | 64 | 63 | 47 | 61 | 59 | 43 | 48 | 65 | 62 | 57 | 61 |
| MPNet | 60 | 61 | 67 | 69 | 66 | 63 | 63 | 70 | 65 | 67 | 62 | 61 | 43 | 60 | 61 | 55 | 49 | 61 | 60 | 65 | 61 | 51 | 63 | 62 | 36 | 51 | 64 | 61 | 63 | 58 |
| XLM-R-dist | 59 | 61 | 67 | 68 | 66 | 62 | 62 | 70 | 65 | 67 | 63 | 61 | 45 | 59 | 62 | 58 | 50 | 62 | 58 | 62 | 62 | 49 | 60 | 60 | 41 | 48 | 62 | 62 | 61 | 56 |
| mT5 | 57 | 60 | 66 | 68 | 65 | 63 | 62 | 70 | 64 | 70 | 64 | 59 | 44 | 53 | 60 | 49 | 44 | 62 | 59 | 53 | 58 | 41 | 60 | 55 | 38 | 54 | 52 | 61 | 52 | 45 |
| mBERT | 55 | 56 | 64 | 66 | 63 | 60 | 58 | 66 | 58 | 63 | 60 | 59 | 43 | 54 | 62 | 57 | 42 | 51 | 55 | 55 | 59 | 40 | 54 | 49 | 44 | 46 | 56 | 56 | 66 | 33 |
| DistilmBERT | 54 | 56 | 63 | 65 | 63 | 60 | 57 | 63 | 57 | 62 | 57 | 57 | 41 | 48 | 61 | 54 | 41 | 54 | 51 | 55 | 61 | 38 | 54 | 53 | 43 | 51 | 53 | 58 | 65 | 36 |
| mUSE-dist | 54 | 55 | 63 | 64 | 64 | 60 | 59 | 63 | 59 | 63 | 55 | 57 | 43 | 51 | 58 | 49 | 43 | 53 | 56 | 55 | 58 | 43 | 54 | 54 | 35 | 46 | 47 | 61 | 54 | 47 |

Figure 1: Detailed comparison of models. Legend: **lang** - averaged by all languages, **ds** - averaged by dataset, **ar** - Arabic, **bg** - Bulgarian, **bs** - Bosnian, **cs** - Czech, **de** - German, **en** - English, **es** - Spanish, **fa** - Persian, **fr** - French, **he** - Hebrew, **hi** - Hindi, **hr** - Croatian, **hu** - Hungarian, **it** - Italian, **ja** - Japanese, **lv** - Latvian, **pl** - Polish, **pt** - Portuguese, **ru** - Russian, **sk** - Slovak, **sl** - Slovenian, **sq** - Albanian, **sr** - Serbian, **sv** - Swedish, **th** - Thai, **ur** - Urdu, **zh** - Chinese.

The hyperparameters of models included in the benchmark are as follows. The hidden size of the model was set to the embedding size of each model when used in HL and FT scenarios. The hidden size of the HB scenario was set to $h = 32$. The learning rate for the HL scenario was $\eta = 1 \times 10^{-3}$, fine-tuning used $\eta = 1 \times 10^{-5}$, and HB scenario used $\eta = 5 \times 10^{-3}$. The batch size in HL and HB scenarios was $b = 200$ and $b = 6$ for fine-tuning. Dropout for HB was $d = 0.5$ and $d = 0.2$ for other scenarios. Training took 5 epochs for fine-tuning and 2 epochs for HL and HB scenarios (beyond 2 epochs most models started to overfit). The models are evaluated using the traditional $F_1$ score computed on three levels: the entire dataset, averaged over all datasets, and the internal dataset.

We performed our experiments using Python 3.9 and PyTorch (1.8.1) (and Tensorflow (2.3.0) for original mUSE). Our experimental setup consists of Intel(R) Xeon(R) CPU E5-2630 v4 @ 2.20GHz and Nvidia Tesla V100 16GB.

Here we present one result available through the benchmarks. Due to space constraints, we move the presentation of other case studies to Appendix B. Figure 1 contains $F_1$ scores of all models fine-tuned on all datasets available for a given language. Interestingly, we see little variance in performance between models for high-resource languages and significant deterioration of performance for low-resource languages. An outlier is the aforementioned Portuguese, where the performance is caused by the lack of data points representing news and reviews. Figure 1 directly shows the usefulness of our benchmark. When designing sentiment classification solutions in Spanish, the choice of model is secondary (models' performances are very similar), and other model features can be considered (such as ease of deployment, cost of inference, and memory requirements). However, if sentiment classification is to be applied to Latvian, the performance difference between models can be as high as 32 pp depending on the choice of the model.

# 5 Related work

## 5.1 Multilingual text representations

One of the foundational discoveries in multilingual presentation learning was the fact that latent vector spaces seemed to encode very similar word relationships across a wide spectrum of languages. Needless to say, first monolingual static word embeddings were quickly followed by multilingual word embeddings which provided the basis for multilingual text representations [68]. A popular solution was to use pre-trained monolingual embeddings, such as `word2vec` [49] and align them via linear transformations using parallel multilingual dictionaries [50]. Another approach advocated for joint learning of multilingual word embeddings on pseudo-bilingual datasets, where tokens in one language would be randomly translated to another language [92].

Static multilingual embeddings were superseded by contextual embeddings produced by language models such as BiLSTM [5] and various Transformers [30, 22, 26, 93, 95]. In the case of these more complex architectures, the multilingual capabilities of language models resulted from specific objective functions used during training. These objectives "pushed" the models toward universal multilingual representations by forcing the models to perform machine translation [5], translation language modeling (TLM) [22, 20], or translation ranking [30, 94].

The evaluation of the quality of multilingual text representation is not trivial. Cross-lingual and multilingual tasks are actively developed to foster the development of multilingual models. Examples of such cross-lingual and multilingual tasks include cross-lingual natural language inference [21], question answering [42], named entity recognition [87, 88] or parallel text extraction [97, 96]. For example, results of comparison of LASER, mBERT, and XLM in tasks of named entity recognition and part-of-speech tagging in zero-shot settings suggests that LASER outperforms the latter methods in the case of knowledge transfer [89]. Another important benchmark is XTREME [34], designed for testing the abilities of cross-lingual transfer across 40 languages and 9 tasks. Despite its massive character, XTREME lacks benchmarking task of sentiment analysis. Also, only mBERT, XLM, XLM-R, and MMTE are used as baseline models.

By far, the most popular pre-trained multilingual language model is Multilingual BERT (mBERT) [26]. It has been used in several cross-lingual studies, for instance, in zero-shot knowledge transfer between Slovenian and Croatian languages [62], exploring code-switching on Spanglish and Hinglish [61], and building hierarchical architecture for zero-shot setting [74]. The properties and limitations of mBERT have been extensively studied. One of the studies showed that mBERT is not learning a joint representation of languages. Rather, it partitions the representation space between languages [79]. The authors used the Projection Weighted Canonical Correlation Analysis (PWCCA) to analyze how translations of the same sentence are represented in mBERT layers. The correlations were stable in pre-trained and fine-tuned models, with the effect being more pronounced in deeper layers of the model. The hierarchical structure induced by correlations was similar to the structure produced by genealogical linguistics.

Another interesting finding was that mBERT encodes language-specific information within the parameter space, and this language component is not removed by fine-tuning [44]. The language-specific component of mBERT can be removed by estimating the centroid of the language (the mean of mBERT embeddings of a given language vocabulary) and subtracting this centroid from representations produced in the language. The existence of the language component has been proven in multiple tasks: language classification, language similarity, sentence retrieval, word alignment, and machine translation.

Several works tested the cross-lingual abilities of mBERT as compared to monolingual models, finding its performance on low-resource languages inferior to monolingual models [91]. One experiment used a bilingual version of mBERT and trained it in multiple configurations to test the influence of the linguistic relationship between the source and the target language, the network architecture, and the input and learning objective [37]. The authors found that structural similarity and depth of a model are the most significant factors behind mBERT's cross-lingual performance. On the other hand, multi-head attention was not particularly important. Regarding the low-resource languages, [91] compared mBERT with baseline models on tasks of named entity recognition, universal part-of-speech tagging, and universal dependency parsing. The authors focused on the comparison between low- and high-resource languages (determined by the size of the Wikipedia dump in each language). mBERT was found to work well on high-resource languages but under-performed on low-resource

languages. Finally, Liu et al. [46] analyzed the relationship between the contextual aspect of mBERT and the training dataset size in the context of multilingual datasets. The authors compared contextual mBERT embeddings with non-contextual models of Word2Vec and GloVe. As it turns out, mBERT outperforms non-contextual models only when large datasets are available for training or fine-tuning. In the low data regime, as well as when the context window is short (limited input), multilingual mBERT is surpassed by static token embeddings.

## 5.2 Multilingual sentiment classification

Several surveys present an overview of traditional sentiment classification methods (lexicon-based approaches and shallow models with lexical features engineering) [25, 70]. Recently, deep-learning approaches became more prevalent. Attia et al. [6] use convolutional neural networks on word-level embeddings of texts in English, German and Arabic. The approach is computationally expensive as it requires separate embedding dictionaries for each language. An alternative approach is to use character-level embeddings. Wehrmann et al. [90] trained such a model for binary sentiment classification of English, German, Portuguese, and Spanish tweets. A similar model was used in [4], but the authors applied multi-stage pre-processing, including dictionary checks, Soundex algorithm, and lemmatization. Other examples of deep neural models for multilingual sentiment classification include recurrent neural networks supplied with machine translation. Can et al. [16] trained a model on English reviews and evaluated it on machine-translated reviews in Russian, Spanish, Turkish, and Dutch using the Google Translation API and pre-trained GloVe embeddings for English. A similar approach is presented in [38], where the authors used LASER sentence embeddings to train a sentiment classifier on Polish reviews and used this classifier to score reviews translated into other languages.

## 5.3 Multilingual sentiment datasets

Multilingual sentiment datasets are crucial resources for training sentiment analysis models that can operate across various languages. Unfortunately, very few multilingual datasets contain sentiment polarity annotations. The Multilingual Amazon Reviews Corpus [57] is a noteworthy example of such a dataset. It provides customer review data in English, Japanese, German, French, Spanish, and Chinese, annotated for sentiment. Another significant resource is the Multilingual Sentiment Lexicons developed by Chen and Skiena [18]), which provides sentiment lexicons for 136 languages. Being a lexicon, it cannot be used to train sentiment models directly, but it can be used for weak supervision. The NTCIR corpus contains information on sentiment polarity for news related to sport and politics for articles in English, Chinese, and Japanese [76]. The JMTC dataset [52](Jigsaw Multilingual Toxic Comment Classification) dataset contains comments from Wikipedia's talk page edits and is available in multiple languages. It is annotated for toxic vs. non-toxic sentiment, which cannot be aligned with our 3-class annotation schema. An interesting resource is Task 4 from SemEval providing English, Spanish, Dutch, Arabic, Russian, and Turkish tweets and SMS messages annotated for 3-class sentiment [54, 67]. The most similar dataset to ours is XED compiled by Öhman et al. [59]. However, this dataset contains original annotations only for English and Finnish, and the remaining 30 languages are "projected" (i.e., annotated samples in English and Finnish are translated into multiple languages).

## 6 Discussion

The main focus of the paper is the introduction of a massively multilingual large collection of sentiment datasets and the extensive benchmark of model training and validation scenarios. During the construction of the benchmark, we have gathered valuable experiences that we want to share with the community. Our most important observation is that a single multilingual sentiment classifier can perform approximately equally well for all languages. A small group of pre-trained models (XML-R, LaBSE, MPNet) under fine-tuning scenario produces the best results relative to other models. Obviously, the absolute $F_1$ score across languages differs significantly (due to data scarcity, data quality, or difficulty of samples), but these selected models always produce top classifiers. Secondly, all models perform better under the fine-tuning scenario, but the performance gain varies from model to model. When evaluated on the test dataset, models gained between 4 pp. (mUSE-dist) and 9 pp. (mBERT, DistilmBERT). When tested on the internal datasets, $F_1$ gains varied between 0 pp (mUSE-

dist) and 20 pp. (DistilmBERT), with mT5 and XML-R improving by 17 pp. and 15 pp., respectively. In general, the largest gains can be observed for models trained with the masked language modeling technique (XML-R, mBERT). We also observe that models which produce lower-quality sentence embeddings gain more from fine-tuning.

We also find that bigger models tend to achieve better performance in all data modalities and training scenarios. Of course, there are counterexamples, e.g., mUSE-dist is smaller than mBERT but achieves better performance in the HL scenario for all datasets. This indicates that the size of the model is an important factor for determining its performance, but other factors, like the domain and the type of pretraining task, may also affect the results. Moreover, we observe that the correlation between model size and model performance is weaker after fine-tuning. This means that one may often find a competitive model with similar performance to the state-of-the-art model but significantly smaller and faster for the production environment.

Our final remark is a warning: in our experiments, we have encountered a significant variability in performance conditioned on a particular data split. In one of the cross-validation fine-tuning experiments, we observed a data fold that produced consistently worse results (up to 4 pp.) for all models. This fold was produced by the same random seed and (most probably) consisted of samples with increased difficulty. Since our experiments involved multiple models, multiple training scenarios, multiple datasets, and multiple languages, we had to rely on data subsampling. Conducting experiments on a fixed seed for sample selection could lead to a similar data fold and, consequently, to biased experimental results.

# 7 Limitations

Despite the fact that our collection is the largest public collection of multilingual sentiment datasets, it still covers only 27 languages. The collection of datasets is highly biased towards the Indo-European family of languages, English in particular. We attribute this bias to the general culture of scientific publishing and its enforcement of English as the primary carrier of scientific discovery. Our work's main potential negative social impact is that the models developed and trained using the provided datasets may still exhibit better performance for the major languages. This could further perpetuate the existing language disparities and inequality in sentiment analysis capabilities across different languages. Addressing this limitation and working towards more equitable representation and performance across languages is crucial to avoid reinforcing language biases and the potential marginalization of underrepresented languages. The ethical implications of such disparities should be thoroughly discussed and considered.

An important limitation of our dataset collection is a significant variance in sample quality across all datasets and all languages. Figure 2 presents the distribution of self-confidence label-quality score for each data point computed by the `cleanlab` [58]. The distribution of quality is skewed in favor of popular languages, with low-resource languages suffering from data quality issues. A related limitation is caused by an unequal distribution of data modalities across languages. For instance, our benchmark clearly shows that all models universally underperform when tested on Portuguese datasets. This is the direct result of the fact that data points for Portuguese almost exclusively represent the domain of social media. As a consequence, some combinations of filtering facets in our dataset collection produce very little data (i.e., asking for social media data in the Germanic genus of Indo-European languages will produce a significantly larger dataset than asking for news data representing Afro-Asiatic languages).

Finally, we acknowledge the lack of internal coherence of annotation protocols between datasets and languages. We have enforced strict quality criteria and rejected all datasets published without the annotation protocol, but we were unable, for obvious reasons, to unify annotation guidelines. The annotation of sentiment expressions and the assignment of sentiment labels are heavily subjective and, at the same time, influenced by cultural and linguistic features. Unfortunately, it is possible that semantically similar utterances will be assigned conflicting labels if they come from different datasets or modalities.

## Acknowledgments and Disclosure of Funding

The work presents results achieved in projects funded by (1) European Regional Development Fund (ERDF) in RPO WD 2014-2020 (project no. RPDS.01.02.01-02-0065/20); (2) the Polish Ministry of Education and Science, CLARIN-PL; (3) the ERDF in the 2014-2020 Smart Growth Operational Programme, CLARIN – Common Language Resources and Technology Infrastructure; (4) project CLARIN-Q (agreement no. 2022/WK/09); (5) the Department of Artificial Intelligence at Wroclaw University of Science and Technology. Mikołaj Morzy is supported by (6) EEA Financial Mechanism 2014-2021 Project 2019/35/J/HS6/03498.

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

# A  Datasets

We present detailed lists of datasets included in our research in Tables 3 and 4. They include language, category, dataset size, class balance, and basic dataset characteristics.

Table 3: List of all monolingual datasets used in experiments. Category (Cat.): R - Reviews, SM - Social Media, C - Chats, N - News, P - Poems, M - Mixed. HL - human labeled, #Words and #Chars are mean values

| Paper | Lang | Cat. | HL | Samples | NEG/NEU/POS | #Words | #Char. |
|---|---|---|---|---|---|---|---|
| Al Omari et al. [1] | ar | R | No | 3096 | 13.0/10.2/76.8 | 9 | 51 |
| Elnagar et al. [29] | ar | R | No | 400101 | 13.0/19.9/67.1 | 22 | 127 |
| Aly and Atiya [2] | ar | R | No | 6250 | 11.6/17.9/70.5 | 65 | 343 |
| Elnagar and Einea [28] | ar | R | No | 504007 | 15.4/21.0/63.6 | 77 | 424 |
| Baly et al. [7] | ar | SM | Yes | 2809 | 47.2/23.9/29.0 | 22 | 130 |
| Nabil et al. [53] | ar | SM | Yes | 3224 | 50.9/25.0/24.1 | 16 | 94 |
| Salameh et al. [71] | ar | SM | Yes | 1199 | 48.0/10.5/41.5 | 11 | 51 |
| Salameh et al. [71] | ar | SM | Yes | 1998 | 67.5/10.1/22.4 | 20 | 107 |
| Habernal et al. [32] | cs | R | No | 91140 | 32.4/33.7/33.9 | 50 | 311 |
| Habernal et al. [32] | cs | R | No | 92758 | 7.9/23.4/68.7 | 20 | 131 |
| Habernal et al. [32] | cs | SM | Yes | 9752 | 20.4/53.1/26.5 | 10 | 59 |
| Habernal et al. [32] | cs | SM | Yes | 2637 | 30.8/60.6/8.6 | 33 | 170 |
| Cieliebak et al. [19] | de | SM | Yes | 9948 | 16.3/59.2/24.6 | 11 | 86 |
| Schabus and Skowron [75] | de | SM | Yes | 3598 | 47.3/51.5/1.2 | 33 | 237 |
| Chapuis et al. [17] | en | C | Yes | 12138 | 31.8/46.5/21.7 | 12 | 48 |
| Chapuis et al. [17] | en | C | Yes | 4643 | 22.3/48.9/28.8 | 15 | 71 |
| Malo et al. [48] | en | N | Yes | 3448 | 12.2/62.1/25.7 | 22 | 124 |
| Bastan et al. [10] | en | N | Yes | 5333 | 11.6/37.3/51.0 | 388 | 2129 |
| Hutto and Gilbert [35] | en | N | No | 5190 | 29.3/52.9/17.8 | 17 | 104 |
| Sheng and Uthus [78] | en | P | Yes | 1052 | 18.3/15.8/65.9 | 7 | 37 |
| Hutto and Gilbert [35] | en | R | No | 3708 | 34.2/19.5/46.3 | 16 | 87 |
| Hutto and Gilbert [35] | en | R | No | 10605 | 49.6/1.5/48.9 | 19 | 111 |
| Ni et al. [56] | en | R | No | 1883238 | 8.3/8.0/83.7 | 70 | 382 |
| Sanders [72] | en | SM | Yes | 3424 | 16.7/68.1/15.2 | 14 | 97 |
| Thelwall et al. [84] | en | SM | Yes | 11759 | 28.0/34.0/38.0 | 26 | 147 |
| Inc. [36] | en | SM | Yes | 14427 | 63.0/21.2/15.8 | 17 | 104 |
| Hutto and Gilbert [35] | en | SM | No | 4200 | 26.9/17.0/56.1 | 13 | 79 |
| Keith Norambuena et al. [39] | es | R | Yes | 382 | 45.0/27.2/27.8 | 165 | 1033 |
| Cruz et al. [23] | es | R | No | 3871 | 32.9/32.3/34.9 | 511 | 3000 |
| Hosseini et al. [33] | fa | R | Yes | 13525 | 12.0/37.5/50.5 | 21 | 104 |
| Amram et al. [3] | he | SM | Yes | 8619 | 26.5/2.8/70.8 | 22 | 110 |
| Pelicon et al. [63] | hr | N | Yes | 2025 | 22.5/61.4/16.0 | 161 | 1021 |
| Barbieri et al. [8] | it | SM | Yes | 8926 | 36.7/41.7/21.6 | 14 | 101 |
| Sprugnoli [82] | it | SM | Yes | 3139 | 24.4/14.9/60.6 | 17 | 106 |
| Sprogis and Rikters [81] | lv | SM | Yes | 5790 | 23.8/45.2/31.0 | 20 | 138 |
| Rybak et al. [69] | pl | R | No | 10074 | 30.8/13.2/56.0 | 80 | 494 |
| Kocoń et al. [41] | pl | R | Yes | 57038 | 42.4/26.8/30.8 | 30 | 175 |
| Sobkowicz and Sobkowicz [80] | pl | SM | Yes | 645 | 50.7/47.3/2.0 | 33 | 230 |
| Brum and Volpe Nunes [13] | pt | SM | Yes | 10109 | 28.8/25.1/46.1 | 12 | 74 |
| Rogers et al. [65] | ru | SM | Yes | 23226 | 16.8/54.6/28.6 | 12 | 79 |
| Bučar et al. [14] | sl | N | Yes | 10417 | 32.0/52.0/16.0 | 309 | 2017 |
| Batanović et al. [11] | sr | R | No | 4724 | 17.8/43.7/38.5 | 498 | 3097 |
| Batanović et al. [12] | sr | R | No | 3948 | 30.3/18.1/51.5 | 18 | 105 |
| Thongthanomkul et al. [86] | th | R | No | 46193 | 5.4/30.5/64.1 | 29 | 544 |
| Suriyawongkul et al. [83] | th | SM | Yes | 26126 | 26.1/55.6/18.3 | 6 | 90 |
| Sharf and Rahman [77] | ur | M | Yes | 19660 | 26.7/43.6/29.7 | 13 | 69 |
| Lin et al. [45] | zh | R | No | 125725 | 28.6/21.9/49.5 | 51 | 128 |

Table 4: List of all multilingual datasets used in experiments. Category (Cat.): R - Reviews, SM - Social Media, C - Chats, N - News, P - Poems, M - Mixed. HL - human labeled

| Paper | Cat. | Lang | HL | Samples | (NEG/NEU/POS) | #Words | #Char. |
|---|---|---|---|---|---|---|---|
| Narr et al. [55] | SM | de | Yes | 953 | 10.0/75.1/14.9 | 12 | 80 |
| | | de | Yes | 1781 | 16.9/63.3/19.8 | 13 | 81 |
| | | en | Yes | 7073 | 17.4/60.0/22.6 | 14 | 78 |
| | | fr | Yes | 685 | 23.4/53.4/23.2 | 14 | 82 |
| | | fr | Yes | 1786 | 25.0/54.3/20.8 | 15 | 83 |
| | | pt | Yes | 759 | 28.1/33.2/38.7 | 14 | 78 |
| | | pt | Yes | 1769 | 30.7/33.9/35.4 | 14 | 78 |
| Keung et al. [40] | R | de | No | 209073 | 40.1/20.0/39.9 | 33 | 208 |
| | | en | No | 209393 | 40.0/20.0/40.0 | 34 | 179 |
| | | es | No | 208127 | 40.2/20.0/39.8 | 27 | 152 |
| | | fr | No | 208160 | 40.2/20.1/39.7 | 28 | 160 |
| | | ja | No | 209780 | 40.0/20.0/40.0 | 2 | 101 |
| | | zh | No | 205977 | 39.8/20.1/40.1 | 1 | 50 |
| Rosenthal et al. [66] | M | ar | Yes | 9391 | 35.5/40.6/23.9 | 14 | 105 |
| | | en | Yes | 65071 | 19.1/45.7/35.2 | 18 | 111 |
| Patwa et al. [60] | SM | es | Yes | 14920 | 16.8/33.1/50.0 | 16 | 86 |
| | | hi | Yes | 16999 | 29.4/37.6/33.0 | 27 | 128 |
| Mozetič et al. [51] | SM | bg | Yes | 62150 | 22.6/45.9/31.5 | 12 | 85 |
| | | bs | Yes | 36183 | 33.4/30.5/36.1 | 12 | 75 |
| | | de | Yes | 90534 | 19.7/52.8/27.4 | 12 | 94 |
| | | en | Yes | 85784 | 26.8/44.1/29.1 | 12 | 77 |
| | | es | Yes | 191412 | 11.8/37.9/50.3 | 14 | 92 |
| | | hr | Yes | 75569 | 25.7/23.9/50.4 | 12 | 91 |
| | | hu | Yes | 56682 | 15.9/31.0/53.1 | 11 | 83 |
| | | pl | Yes | 168931 | 30.0/26.1/43.9 | 11 | 82 |
| | | pt | Yes | 145197 | 37.2/35.0/27.8 | 10 | 61 |
| | | ru | Yes | 87704 | 32.0/40.1/27.8 | 10 | 67 |
| | | sk | Yes | 56623 | 25.6/22.5/51.9 | 13 | 97 |
| | | sl | Yes | 103126 | 29.9/43.3/26.8 | 13 | 91 |
| | | sq | Yes | 44284 | 15.7/33.1/51.1 | 13 | 90 |
| | | sr | Yes | 67696 | 34.8/42.8/22.4 | 13 | 81 |
| | | sv | Yes | 41346 | 40.3/31.2/28.5 | 14 | 94 |

# B Experiments

## B.1 Data quality

In this section, we expand upon the results of experiments included in the benchmark. We begin with the presentation of the distribution of data point quality across all datasets. The data point quality has been computed using the `cleanlab` library self-confidence label-quality score [58]. Figure 2 presents the cumulative distribution of data quality score. As we can see, the data quality still is a pressing issue, despite our efforts to publish a well-curated collection. Over 40% of all data points have quality 0.6 or worse. As we have mentioned, low-quality data points are not distributed uniformly across all languages and datasets, but the impact of low-quality data is most pronounced for low-resource languages.

## B.2 Pairwise ranked comparison of models

Of course, everyone is most interested in finding the answer to the following question: "Which model should I use?" The answer, as usual, is: "It depends". Figure 3 presents the results of the Nemenyi pairwise rank comparison test. In neither of the considered training scenarios (HL, HB, TF), a single model is statistically better than others. However, these results allow us to draw partial conclusions. For instance, the MPNet performs best in the HL scenario, and the same model is not significantly worse than XML-M, which is the best model in HB and TF scenarios. We can also notice that mBERT-based models (mBERT and DistillmBERT) proved to be the worst language models for our

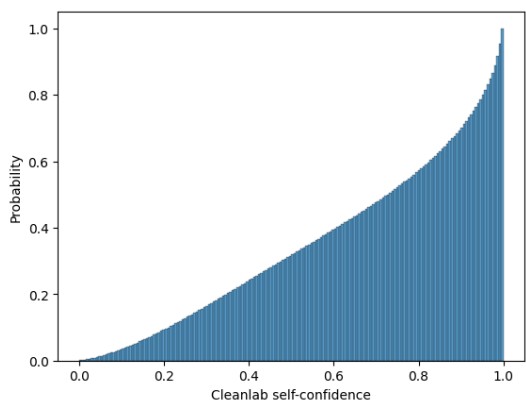

Figure 2: Cumulative distribution of sample quality across all datasets

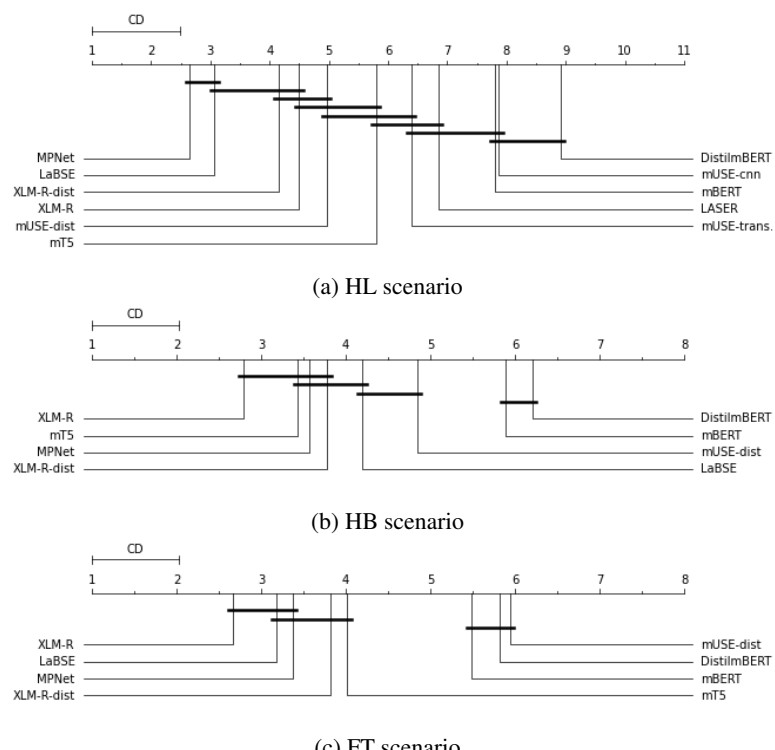

(a) HL scenario

(b) HB scenario

(c) FT scenario

Figure 3: Nemenyi diagrams of model ranks according to the F1-score on each dataset

tasks. Three models stand out as the most promising choices: XLM-R, LaBSE, and MPNet. They achieve comparable performance in all scenarios and test cases. Furthermore, they are better than other models in almost all test cases.

Table 5: Aggregated $F_1$ results of models. The best results for each test set are highlighted. W - whole test, A - avg. by dataset, I - internal

| | XLM-R | LaBSE | MPNet | XLM-R-dist | mT5 | mBERT | DistilmBERT | mUSE-dist | LASER | mUSE-trans. | mUSE-cnn |
|---|---|---|---|---|---|---|---|---|---|---|---|
| | | | | | HL scenario | | | | | | |
| W | 62 | 62 | **63** | 60 | 59 | 56 | 55 | 59 | 55 | 55 | 54 |
| A | 51 | 54 | **55** | 51 | 49 | 45 | 43 | 50 | 47 | 47 | 45 |
| I | 55 | **61** | **61** | 56 | 50 | 43 | 38 | 60 | 50 | 49 | 50 |
| | | | | | HB scenario | | | | | | |
| W | **66** | 62 | 63 | 62 | 65 | 60 | 59 | 62 | - | - | - |
| A | **57** | 55 | 56 | 54 | 56 | 49 | 48 | 54 | - | - | - |
| I | **64** | 63 | **64** | 63 | 63 | 54 | 48 | **64** | - | - | - |
| | | | | | FT scenario | | | | | | |
| W | **68** | **68** | 67 | 67 | 66 | 65 | **64** | 63 | - | - | - |
| A | 61 | **62** | **62** | **62** | 60 | 56 | 56 | 56 | - | - | - |
| I | **70** | 69 | 65 | 67 | 67 | 57 | 58 | 60 | - | - | - |

## B.3 Detailed results for each model

Table 5 shows the $F_1$ scores of all models aggregated by datasets. The models improve with fine-tuning (up to 0.7 $F_1$) as compared to linear (0.61) or BiLSTM (0.64) layers operating on embeddings produced by models. The performance gains are more pronounced for models trained with masked language modeling objectives (mBERT, XML-R) than for models trained with sentence classification or sentence similarity objectives (LaBSE). Fine-tuning reduces inequalities in the results between models (0.55 vs. 0.43 for best and worst models in the HL scenario). The additional BiLSTM layer on top of transformer-based token embeddings has the capacity to improve the results compared to the linear classification layer model. The differences are most clear in the case of the results for our internal dataset, where the result improved even by 13 pp. for the mT5 model.

## B.4 Detailed results for each language

We assessed the performance of each model in each experimental scenario per language. The texts were sub-sampled with stratification by language and class label so that language distribution in the test dataset follows the distribution in the whole dataset. We also include the total macro $F_1$ score value in column "all". Results are presented in Figure 4. Those results confirm our previous findings regarding the advantage of XLM-R, LaBSE, and MPNet models. They outperform other models in most languages, with no clear indication of superiority among them.

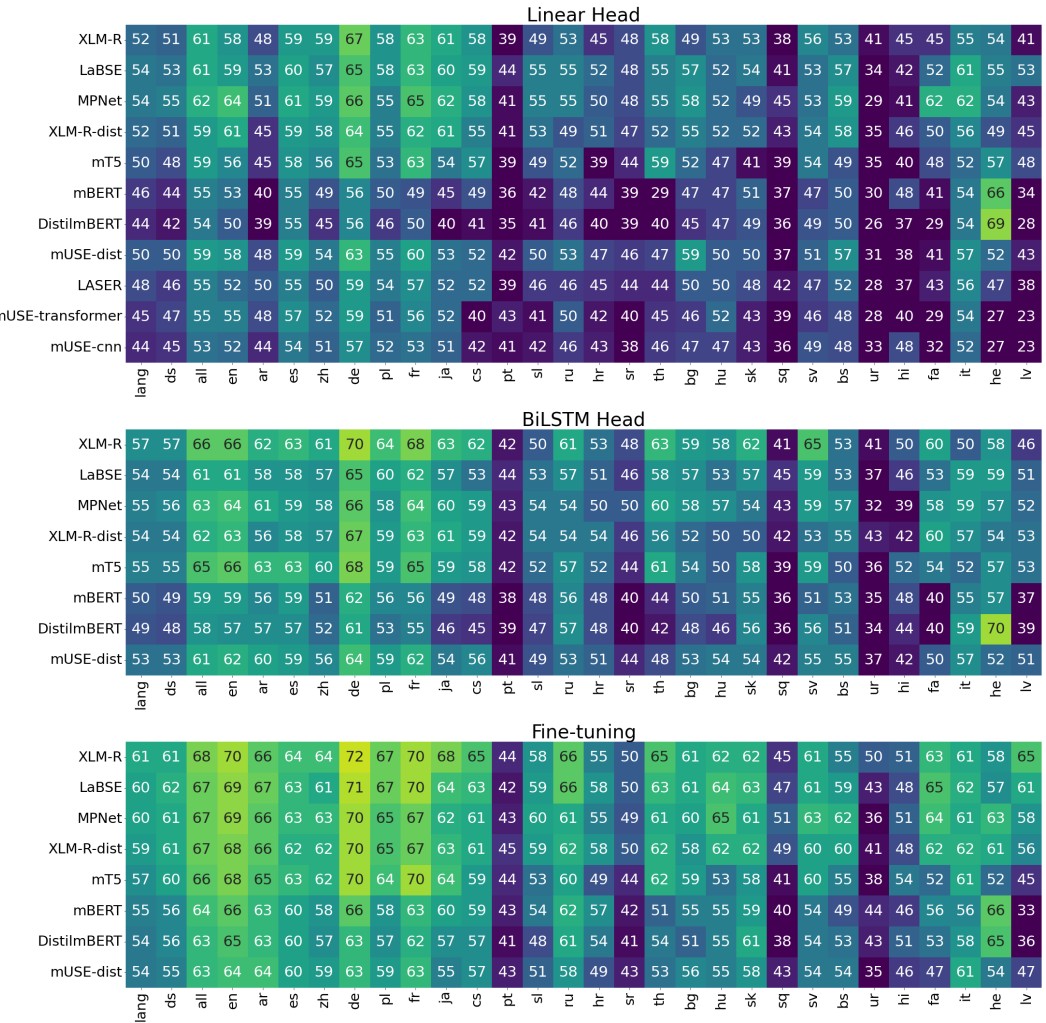

Figure 4: Detailed results of models' comparison. Legend: lang - averaged by all languages, ds - averaged by dataset, ar - Arabic, bg - Bulgarian, bs - Bosnian, cs - Czech, de - German, en - English, es - Spanish, fa - Persian, fr - French, he - Hebrew, hi - Hindi, hr - Croatian, hu - Hungarian, it - Italian, ja - Japanese, lv - Latvian, pl - Polish, pt - Portuguese, ru - Russian, sk - Slovak, sl - Slovenian, sq - Albanian, sr - Serbian, sv - Swedish, th - Thai, ur - Urdu, zh - Chinese.

## B.5 Transfer between data modalities

The last example compares the effectiveness of transfer learning between data modalities. The results are presented in Figure 5. As expected, when models are tested against datasets from the same domain (news, social media, reviews), the average performance is much higher than on out-of-domain datasets. What is striking is the visible deterioration of the performance of models trained in the news domain. When knowledge transfer happens between social media and review domains, the average performance of models stays relatively the same.

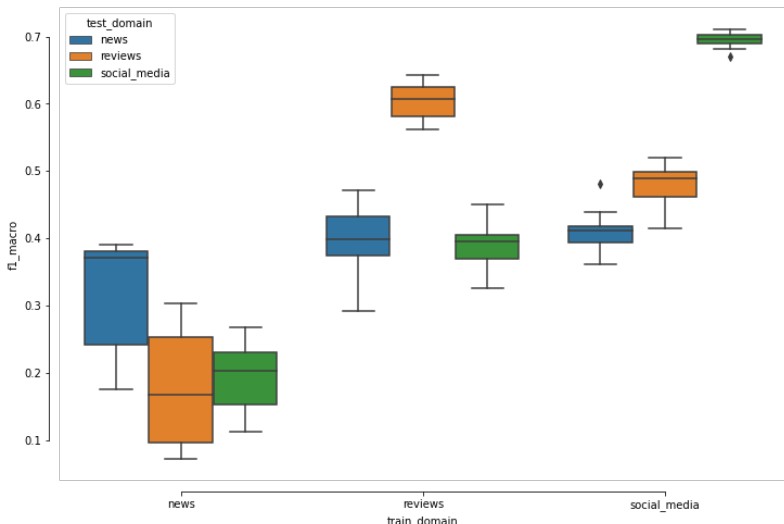

Figure 5: Transfer learning between data modalities

## C Listings

The ease of loading and using more than 6M sentiment annotated texts is as simple as the listings above. All data will be downloaded automatically; you must only appropriately filter it to your use case.

```python
import datasets

mms_dataset = datasets.load_dataset("Brand24/mms")
slavic = mms_dataset.filter(lambda row: row["Genus"] == "Slavic" and
    row["Polar questions"] == "interrogative word order")
```
Listing 1: Loading and filtering sentiment examples for Slavic *genus* with polar questions formed using interrogative word order.

```python
import datasets

mms_dataset = datasets.load_dataset("Brand24/mms")
afro_asiatic = mms_dataset.filter(lambda row: row["Family"] == "Afro-
    Asiatic" and row["Number of cases"] == "no morphological case-
    making")
```
Listing 2: Loading and filtering sentiment examples for Afro-Asiatic Language Family with no morphological case-making.

```python
import datasets

mms_dataset = datasets.load_dataset("Brand24/mms")
pl_only_social_media = mms_dataset.filter(lambda row: row['language']
    == 'pl' and row['domain'] == "social_media")
```
Listing 3: Loading and filtering sentiment examples for specific language and domain

```python
import datasets

CLEANLAB_THRESHOLD = 0.7

mms_dataset = datasets.load_dataset("Brand24/mms")
afro_asiatic = mms_dataset.filter(lambda row: row["
    cleanlab_self_confidence"] >= CLEANLAB_THRESHOLD)
```
Listing 4: Loading data based on cleanlab data quality score

# D Error analysis

We present the error analysis of sentiment model results. The Tables 6 and 7 could be a little bit small, so we advise checking the editable version[3] of results with the option to check ours or create your pivot tables of the results.

A couple of patterns could be spotted in Table 6.

- Japanese neutral class is consistently low performing across all models, not exceeding even 50% f1 score. Interestingly, even well-resourced languages like English still have problems with the neutral class. only one English model **FT_XLM-R** achieved 0.7 F1 score.
- The positive class is the easiest class to learn for the models, but the performance of neutrals and negatives varies very much, and it always needs to be verified, and potentially there could be a need to annotate more data for these classes.

Table 6: F1 scores across all languages and sentiment classes. Legend: ar - Arabic, bg - Bulgarian, bs - Bosnian, cs - Czech, de - German, en - English, es - Spanish, fa - Persian, fr - French, he - Hebrew, hi - Hindi, hr - Croatian, hu - Hungarian, it - Italian, ja - Japanese, lv - Latvian, pl - Polish, pt - Portuguese, ru - Russian, sk - Slovak, sl - Slovenian, sq - Albanian, sr - Serbian, sv - Swedish, th - Thai, ur - Urdu, zh - Chinese.

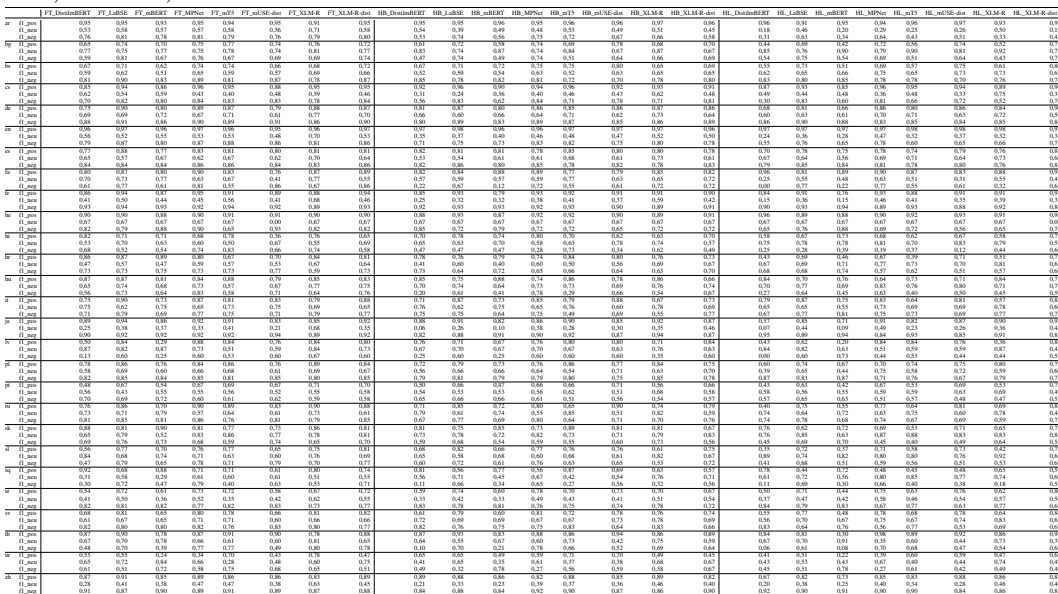

A couple of patterns could be spotted in Table 7.

- The first interesting pattern is a hard-to-learn neutral class for 'poems'. It could be related to a tiny dataset, only with approximately 1000 English examples (693 positive, 193 negative, and 166 neutral examples).
- The review texts are easy to train for positive and negative; however, they are tough to work decently for neutral class. The reason for it could be mixing positive and negative sentiment in many neutral annotated reviews, mentioning both advantages and disadvantages of products or services.
- News domain shows interestingly high performance for the neutral class; however, it could be related to skewed distribution in this domain for the neutral class. We have neutral as many positive and negative examples summed together.

There are many other angles on how we can use our benchmarking results. We advise you to check the editable results and create your pivot tables for your own purposes and needs.

---

[3] https://drive.google.com/drive/folders/1JKBlslTXXiTVu9cHhQL7Fo4GdhUmdInx?usp=sharing

Table 7: F1 scores across domains. Legend: ar - Arabic, bg - Bulgarian, bs - Bosnian, cs - Czech, de - German, en - English, es - Spanish, fa - Persian, fr - French, he - Hebrew, hi - Hindi, hr - Croatian, hu - Hungarian, it - Italian, ja - Japanese, lv - Latvian, pl - Polish, pt - Portuguese, ru - Russian, sk - Slovak, sl - Slovenian, sq - Albanian, sr - Serbian, sv - Swedish, th - Thai, ur - Urdu, zh - Chinese.

| | chats | | | mixed | | | news | | | poems | | | reviews | | | social_media | | |
|---|---|---|---|---|---|---|---|---|---|---|---|---|---|---|---|---|---|---|
| | f1_pos | f1_neu | f1_neg | f1_pos | f1_neu | f1_neg | f1_pos | f1_neu | f1_neg | f1_pos | f1_neu | f1_neg | f1_pos | f1_neu | f1_neg | f1_pos | f1_neu | f1_neg |
| FT_DistilmBERT | 0,70 | 0,73 | 0,48 | 0,75 | 0,72 | 0,65 | 0,61 | 0,86 | 0,50 | 0,73 | 0,00 | 0,67 | 0,94 | 0,47 | 0,86 | 0,74 | 0,68 | 0,70 |
| FT_LaBSE | 0,66 | 0,59 | 0,65 | 0,80 | 0,73 | 0,73 | 0,76 | 0,82 | 0,77 | 0,60 | 0,00 | 0,67 | 0,96 | 0,50 | 0,89 | 0,81 | 0,66 | 0,76 |
| FT_mBERT | 0,66 | 0,69 | 0,55 | 0,67 | 0,80 | 0,69 | 0,66 | 0,78 | 0,63 | 0,83 | 0,00 | 0,67 | 0,94 | 0,50 | 0,86 | 0,74 | 0,68 | 0,73 |
| FT_MPNet | 0,72 | 0,62 | 0,67 | 0,78 | 0,74 | 0,74 | 0,61 | 0,78 | 0,80 | 0,92 | 0,00 | 1,00 | 0,96 | 0,48 | 0,89 | 0,78 | 0,68 | 0,76 |
| FT_mT5 | 0,70 | 0,47 | 0,70 | 0,79 | 0,73 | 0,73 | 0,74 | 0,87 | 0,57 | 0,92 | 0,00 | 0,67 | 0,95 | 0,51 | 0,89 | 0,78 | 0,66 | 0,71 |
| FT_mUSE-dist | 0,66 | 0,66 | 0,55 | 0,69 | 0,73 | 0,70 | 0,41 | 0,79 | 0,70 | 0,73 | 0,67 | 1,00 | 0,94 | 0,43 | 0,88 | 0,73 | 0,64 | 0,75 |
| FT_XLM-R | 0,68 | 0,71 | 0,63 | 0,78 | 0,77 | 0,68 | 0,64 | 0,86 | 0,70 | 0,73 | 0,00 | 1,00 | 0,93 | 0,67 | 0,86 | 0,80 | 0,72 | 0,69 |
| FT_XLM-R-dist | 0,70 | 0,77 | 0,70 | 0,77 | 0,77 | 0,71 | 0,51 | 0,82 | 0,74 | 0,60 | 0,00 | 1,00 | 0,96 | 0,47 | 0,89 | 0,78 | 0,69 | 0,74 |
| HB_DistilmBERT | 0,80 | 0,42 | 0,34 | 0,74 | 0,68 | 0,42 | 0,59 | 0,57 | 0,52 | 0,73 | 0,00 | 0,67 | 0,95 | 0,29 | 0,78 | 0,73 | 0,69 | 0,64 |
| HB_LaBSE | 0,70 | 0,67 | 0,67 | 0,81 | 0,72 | 0,59 | 0,75 | 0,82 | 0,45 | 0,73 | 0,00 | 1,00 | 0,96 | 0,30 | 0,84 | 0,75 | 0,64 | 0,73 |
| HB_mBERT | 0,84 | 0,36 | 0,48 | 0,71 | 0,68 | 0,56 | 0,55 | 0,78 | 0,59 | 0,73 | 0,00 | 0,67 | 0,94 | 0,34 | 0,81 | 0,73 | 0,68 | 0,63 |
| HB_MPNet | 0,66 | 0,67 | 0,57 | 0,76 | 0,70 | 0,67 | 0,57 | 0,74 | 0,67 | 0,83 | 0,00 | 1,00 | 0,95 | 0,40 | 0,87 | 0,73 | 0,65 | 0,73 |
| HB_mT5 | 0,78 | 0,46 | 0,57 | 0,76 | 0,72 | 0,58 | 0,78 | 0,85 | 0,43 | 0,92 | 0,00 | 0,67 | 0,96 | 0,43 | 0,86 | 0,78 | 0,71 | 0,62 |
| HB_mUSE-dist | 0,76 | 0,63 | 0,45 | 0,74 | 0,74 | 0,58 | 0,63 | 0,84 | 0,45 | 0,60 | 0,00 | 0,67 | 0,95 | 0,39 | 0,82 | 0,76 | 0,64 | 0,65 |
| HB_XLM-R | 0,74 | 0,66 | 0,57 | 0,68 | 0,82 | 0,59 | 0,57 | 0,83 | 0,55 | 0,92 | 0,00 | 0,00 | 0,96 | 0,47 | 0,84 | 0,73 | 0,77 | 0,65 |
| HB_XLM-R-dist | 0,63 | 0,61 | 0,65 | 0,70 | 0,73 | 0,68 | 0,53 | 0,78 | 0,52 | 0,60 | 0,00 | 1,00 | 0,95 | 0,43 | 0,82 | 0,71 | 0,67 | 0,69 |
| HL_DistilmBERT | 0,84 | 0,34 | 0,52 | 0,78 | 0,57 | 0,43 | 0,64 | 0,76 | 0,29 | 0,92 | 0,00 | 0,00 | 0,93 | 0,17 | 0,74 | 0,61 | 0,70 | 0,63 |
| HL_LaBSE | 0,66 | 0,58 | 0,72 | 0,81 | 0,64 | 0,68 | 0,61 | 0,88 | 0,43 | 0,73 | 0,00 | 1,00 | 0,94 | 0,38 | 0,83 | 0,70 | 0,68 | 0,72 |
| HL_mBERT | 0,85 | 0,32 | 0,43 | 0,76 | 0,59 | 0,55 | 0,59 | 0,75 | 0,37 | 1,00 | 0,00 | 0,67 | 0,92 | 0,21 | 0,79 | 0,62 | 0,69 | 0,66 |
| HL_MPNet | 0,68 | 0,58 | 0,65 | 0,76 | 0,74 | 0,60 | 0,80 | 0,82 | 0,43 | 0,73 | 0,00 | 1,00 | 0,96 | 0,36 | 0,82 | 0,68 | 0,74 | 0,63 |
| HL_mT5 | 0,78 | 0,29 | 0,50 | 0,80 | 0,69 | 0,53 | 0,79 | 0,76 | 0,55 | 0,92 | 0,00 | 0,67 | 0,96 | 0,29 | 0,77 | 0,62 | 0,73 | 0,61 |
| HL_mUSE-dist | 0,78 | 0,67 | 0,52 | 0,81 | 0,67 | 0,48 | 0,75 | 0,73 | 0,48 | 0,83 | 0,00 | 1,00 | 0,96 | 0,26 | 0,76 | 0,73 | 0,72 | 0,56 |
| HL_XLM-R | 0,84 | 0,46 | 0,52 | 0,80 | 0,71 | 0,54 | 0,74 | 0,82 | 0,50 | 0,92 | 0,00 | 0,00 | 0,96 | 0,41 | 0,75 | 0,65 | 0,78 | 0,60 |
| HL_XLM-R-dist | 0,74 | 0,51 | 0,78 | 0,80 | 0,63 | 0,65 | 0,71 | 0,70 | 0,57 | 0,60 | 0,00 | 1,00 | 0,96 | 0,29 | 0,78 | 0,75 | 0,63 | 0,67 |

