# OpenReview forum: "Massively Multilingual Corpus of Sentiment Datasets and Multi-faceted Sentiment Classification Benchmark"
_NeurIPS.cc/2023/Track/Datasets_and_Benchmarks — NeurIPS 2023 Datasets and Benchmarks Poster_

### Official Review · Reviewer_NixT · 2023-07-20
**Review of the largest sentiment analysis benchmark**

**Rating:** 6
**Confidence:** 3

**Strengths:**

The dataset is a valuable resource, which will facilitate research for resource-lean languages.

**Additional Feedback:**

L21: Please explain the translation and the sentiment. A table with all the scores and explanations of the sentiment could assist the reader.

L38: Why isn’t subjectivity (e.g., based on demographics) included here? It is discussed in related work, coupled with low agreement.

L43: FTing yields SOTA, but how is “training” beneficial?

L46/L50: TL is only valid for FT, right? And I guess mainly for multilinguality. Please elaborate.

L131: Please report these 345 datasets.

L143: What is your motivation behind this aggregation strategy?

L145: The NONE class can be inferred when the score is below a threshold for all emotions. Please provide your experimental analysis.

L151: There is work suggesting otherwise (e.g., https://le-wi-di.github.io/, https://link.springer.com/article/10.1007/s12559-022-10088-2). Can you please change the dataset so that the unaggregated labels are also provided?

L160: What is this self-confidence label-quality score? An explanation should be provided along with the citation.

L164-165: Why these two languages?

L164-165: What is the background and any demographics of the annotators?

L164-165: What are the exact guidelines the annotators were given?

L164-165: What is majority labeling during annotation?

L164-165: How was Cohen’s kappa computed among three (not two) annotators?

L188: How were the hyperparameters selected and why weren’t these language-specific?

Figure 2: Both Figure 2 and Figure 1 are written. Please explain also the color scheme.

L261: language mistakes (on the tasks, the authors of [80])

L279: What is the soundex algorithm?

L786: "In neither of the considered training scenarios (HL, HB, TF), a single model is statistically better than others.” Why isn’t this discussed in the main paper?

**Clarity:**

The paper is well written, but there are minor edits that should be done (e.g., Figure 2, which also writes Figure 1).

**Correctness:**

* Given that this work is expected to support "low-resource languages in perfomant sentiment classification", more details should be provided regarding the selection process. To give an example, the datasets described in these papers are available but not included here, leaving out completely the Greek language:
https://dl.acm.org/doi/10.1145/2801948.2802010
https://aclanthology.org/2022.lrec-1.765

**Documentation:**

I might have missed this, but I couldn't find the original pool of 345 datasets used to form the suggested corpus. I suggest that the authors release this information, along with exact details regarding which dataset (language) was removed in each step.

**Ethics:**

My only two concerns (agreement, selection process), discussed in correctness and in opportunities for improvement, can be addressed.

**Limitations:**

Please read my comment in correctness.

**Opportunities For Improvement:**

* The inter-annotator agreement study should be described in details, including releasing the annotator guidelines, discussing the background of the annotators, the kappa computation (not clear now, please see my comments). The authors could also report, and perhaps compare with, the agreement of the original studies the datasets come from.

**Relation To Prior Work:**

This work is clearly different from previous contributions, but the respective section 5.2 might fall short. Scholar, for example, brings this well-cited paper (https://journals.plos.org/plosone/article?id=10.1371/journal.pone.0155036) as the top result for the query: "multilingual sentiment classification".

**Summary And Contributions:**

This study collected 345 datasets for sentiment analysis from multiple languages. Out of these, 79 were selected to form a multilingual dataset and present a benchmark. This work will assist sentiment analysis, especially for resource-lean languages. I have two main criticisms, both of which can be addressed. First, inter-annotator agreement needs to be analysed more. Second, more details should be added regarding the selection process and explain why existing datasets are not considered (leaving out languages, such as Greek).

---

> ### Author Response · Authors · 2023-08-09
> **Addressing Reviewer Feedback: Enhancing our Research Paper with Detailed Dataset Description**
>
> We sincerely thank you for your thorough review and constructive feedback on our paper. Your expertise and the effort you put into this process are truly appreciated. We fully acknowledge that the part of the paper discussing the dataset selection process, selection criteria, annotator agreement, and decisions for dataset rejections must be adequately described. We already started to work on the amendment, which should address most (if not all) of your concerns. As for the detailed comments and questions in the "Additional feedback" section, we will try to include as many as possible in the revised text. Independently, we will provide detailed responses on this forum. We aim to submit the revised version of the paper and provide complete responses to all your questions/comments by August 21, hopefully giving you enough time to evaluate the revision and read our responses. Once again, thank you for your time and valuable input.

---

> ### Author Response · Authors · 2023-08-22
> **Responses to Review of the largest sentiment analysis benchmark**
>
> Once again, thank you for a constructive review.
>
> We will link our responses to each of your comments for easier readability.
>
> Opportunities For Improvement:
>
> `Q: The inter-annotator agreement study should be described in details, including releasing the annotator guidelines, discussing the background of the annotators, the kappa computation (not clear now, please see my comments).`
>
> Response: We added annotation guidelines to the supplementary materials. We also described the IAA metrics and how they were calculated - please see Section 3.2 “Pre-processing of datasets” in the second paragraph. The annotators were two women and one man, all Polish nationals, age 21-25, studying, two annotators with majors in technical sciences, and one majoring in humanities. The annotators were never asked about personal preferences (religion, political leanings, sexual orientation, etc.) but were explicitly asked never to consider such preferences during the annotation process.
>
> `Q: The authors could also report, and perhaps compare with, the agreement of the original studies the datasets come from.`
>
> Response: We didn’t reannotate already annotated datasets; we annotated only our internal dataset; hence it would be tricky and not so objective to compare IAA for different datasets. The annotations guideline, the domain/language of data, and the annotator's team would be different. However, it would be a good idea to plan and conduct some follow-up analysis of already annotated datasets in the future.
>
> `Q: Given that this work is expected to support "low-resource languages in perfomant sentiment classification", more details should be provided regarding the selection process. To give an example, the datasets described in these papers are available but not included here, leaving out completely the Greek language: https://dl.acm.org/doi/10.1145/2801948.2802010 https://aclanthology.org/2022.lrec-1.765`
>
> Response: We answered it in response to all reviewers to not duplicate answer for each of reviewer. In addition, we have already added the first Greek dataset, which could be easily used with this code snippet
>
> ```python
> import datasets
> mms_dataset = datasets.load_dataset("Brand24/mms", download_mode="force_redownload") # it case you have already downloaded the corpus we must refresh HF cache
> greek = mms_dataset.filter(lambda row: row['language'] == 'el')
> ```
>
> `Q: The paper is well written, but there are minor edits that should be done (e.g., Figure 2, which also writes Figure 1).`
>
> Response: Thank you very much for spotting it. We fixed the figure caption.
>
> Relation To Prior Work:
> `Q: This work is clearly different from previous contributions, but the respective section 5.2 might fall short. Scholar, for example, brings this well-cited paper (https://journals.plos.org/plosone/article?id=10.1371/journal.pone.0155036) as the top result for the query: "multilingual sentiment classification".`
>
> Response: Thank you for mentioning the paper, we added Section 5.3 "Multilingual sentiment datasets". We want also to mention that the Mozetic et al. paper is already in Table 3 in the appendix.
>
> `Q: I might have missed this, but I couldn't find the original pool of 345 datasets used to form the suggested corpus. I suggest that the authors release this information, along with exact details regarding which dataset (language) was removed in each step.`
>
> Response: We answered it in response to all reviewers to not duplicate answers for each of reviewer.

---

> > ### Author Response · Authors · 2023-08-22
> > **Response to additional comment**
> >
> > Additional Feedback:
> >
> > `L21: Please explain the translation and the sentiment. A table with all the scores and explanations of the sentiment could assist the reader.`
> > Response: This paragraph serves as a light-hearted introduction to the subject. The phrases are translated verbatim (e.g., the Polish sentence “hotel jak hotel, mogło być gorzej” translates directly into English “hotel like a hotel, all in all, it could have been worse”). All sentiment scores are stated and produced by the twitter-xml-roberta-base model mentioned in the footnote. After reading this paragraph several times, we have decided it is sufficiently clear. It could be improved, as you suggested, but it is not crucial for the paper, and we are already operating a tight space limit. We have decided to leave the paragraph as is.
> >
> > `L38: Why isn’t subjectivity (e.g., based on demographics) included here? It is discussed in related work, coupled with low agreement.`
> > Response: This information is mostly missing from source datasets we compile into the final dataset. Cultural expressions of sentiment can be lost in translation, but a deep dive into this subject is definitely outside the scope of our work. For better or for worse, we had to assume that there is a universal understanding of positive, negative, and neutral utterances (even if it is culture dependent). Also, this is why we don’t use automatically translated datasets since machine translation may skew the sentiment expression, and we rely on datasets produced in native languages.
> >
> > `L43: FTing yields SOTA, but how is “training” beneficial?`
> > Response: The main difference between training and fine-tuning is that fine-tuning uses a model trained on a much larger corpus of data; hence, the base model has a much richer representation of language. Subsequent fine-tuning shifts the parameters associated with common sentiment markers in a language. Training from scratch is much more expensive and requires the compilation of the training dataset, but in certain scenarios, it can be beneficial. For instance, the business task may involve sentiment classification of texts not well represented by the general language model, such as tweets or SMS - a much smaller model trained only on tweets/SMS may outperform the large fine-tuned model.
> >
> > `L46/L50: TL is only valid for FT, right? And I guess mainly for multilinguality. Please elaborate.`
> > Response: This might not be expressed clearly enough in the text. Given a multilingual model M, if we fine-tune this model using an additional sentiment dataset in Polish, just by the sheer similarity of tokens, the performance of model M should increase not only for inputs in Polish but in Czech, Slovak, Slovene, Ukrainian, etc. as well. The transfer learning here is the by-product of fine-tuning the model in a single language. Still, the fine-tuning results are visible for other similar languages (albeit not as pronounced as for the language for which model M has been fine-tuned, obviously). We have added this explanation to the paper.
> >
> > `L131: Please report these 345 datasets.`
> > Response: All the datasets with links to sources/papers have been added as a spreadsheet in the Supplementary Materials section.
> >
> > `L143: What is your motivation behind this aggregation strategy?`
> > Response: This strategy seems to be the most “natural” mapping from a 5-point Likert scale to 3 sentiment classes. We have seen this aggregation in several sentiment datasets, e.g., in SentiNews Dataset (https://www.clarin.si/repository/xmlui/handle/11356/1495) or CMU-MOSI dataset (https://arxiv.org/pdf/1606.06259.pdf)
> >
> > `L145: The NONE class can be inferred when the score is below a threshold for all emotions. Please provide your experimental analysis.`
> > Response: This is undoubtedly true, but unfortunately, most datasets did not report on the confidence scores of labels, just the labels themselves. From this data, there was no way to deduce the neutral class meaningfully. Also, when the annotation task is defined initially as a three-class problem, the annotators tend to move explicit expressions of sentiment to positive/negative classes and put all ambiguous samples in the neutral class. When the dataset has been annotated using the binary labeling scheme, the expressiveness of positive/negative labels close to the decision boundary may be weak. We have decided to reject binary datasets because we did not develop a robust way of ensuring the coherence of annotations.

---

> > > ### Author Response · Authors · 2023-08-22
> > > **Response to additional comments part 2**
> > >
> > > `L160: What is this self-confidence label-quality score? An explanation should be provided along with the citation.`
> > > Response: The holdout probability of a label in the Cleanlab library is the probability that a sample with that label would be classified as positive by a classifier trained on a holdout dataset. It is calculated by averaging the holdout probabilities of all samples with that label. The holdout probability is a useful metric for evaluating the uncertainty of a label. A low holdout probability indicates that a classifier will likely misclassify the label. In contrast, a high holdout probability indicates that the label will likely be correctly classified.
> > >
> > > `L164-165: Why these two languages?`
> > > Response: Three annotators involved in curating the baseline dataset were Polish nationals with fluent knowledge of English. For these two languages, we can be sure that the annotators can precisely determine the sentiment of the utterance. Also, given the business ramifications of Brand24, these two languages are the most important for the company, and annotators have the most expertise and experience annotating these languages.
> > >
> > > `L164-165: What is the background and any demographics of the annotators?`
> > > Response: The annotators were two women and one man, all Polish nationals, age 21-25, studying, two annotators with majors in technical sciences, and one majoring in humanities. The annotators were never asked about personal preferences (religion, political leanings, sexual orientation, etc.) but were explicitly asked never to consider such preferences during the annotation process.
> > >
> > > `L164-165: What are the exact guidelines the annotators were given?`
> > > Response: Detailed annotation guidelines followed during the curation of the baseline dataset are attached in the Supplementary Materials section.
> > >
> > > `L164-165: What is majority labeling during annotation?`
> > > Response: Since three annotators have annotated each text snippet, we accepted the snippet only if at least two annotators agreed on a label. The snippet was rejected from the baseline dataset if three different labels were assigned.
> > > In the paper, we also mention majority-label voting. When verifying datasets, we cross-examined them and found several duplicates of text samples. We have used the majority label to establish a single label after de-duplication (and rejected samples for which no majority label was present)
> > >
> > > `L164-165: How was Cohen’s kappa computed among three (not two) annotators?`
> > > Response: We have used the implementation provided in the nltk.metrics.agreement (https://www.nltk.org/_modules/nltk/metrics/agreement.html), where Cohen’s kappa for n>2 annotators is computed as the average Cohen’s kappa over all pairs of annotators.
> > >
> > > `L188: How were the hyperparameters selected and why weren’t these language-specific?`
> > > Response: We use pre-trained versions of models and report their efficiency in three different scenarios: HL (the model only generates text embeddings), HB (very similar, but the linear classifier working on text embeddings is replaced with BiLSTM), and FT (fine-tuning on sentiment task). Out of these three, only FT requires updating of model weights (based on the training on sentiment datasets). Still, even here, we don’t modify the models’ hyperparameters and use the values described in L.199. These values represent an empirical compromise between the quality of models and the budget for computational experiments. Please note that we had over 30 basic configurations (11 models * 3 deployment strategies), not to mention alternative dataset splits. Hyperparameter tuning for each language was simply unfeasible because all models used in the benchmark are inherently multilingual; fine-tuning on English/German/Spanish might result in visible differences, but the amount of data we have for other languages is not sufficient for a significant shift of the model. In other words, we might have fine-tuned a large multilingual model with data for a specific language, but we would hardly see any tangible results given the amount of data we have.
> > > We want to highlight that our corpus enables anybody to easily slice the dataset and use them in training for more sophisticated experiments with more hyperparameter search spaces. This is one of the most important contributions with our work that the NLP community can accelerate model testing and experimentation with sentiment models using unified layers and easy dataset acquisition and slicing and dicing.
> > >
> > > `Figure 2: Both Figure 2 and Figure 1 are written. Please explain also the color scheme.`
> > > Response: We have mistakenly used two subsequent \caption{} environments; thanks for pointing this out. It was challenging to come up with a reasonable visualization; the figure is a heat map with lighter colors representing better results. If accepted for publication, we will try to improve the quality of the figure and make it more friendly for print-outs.

---

> > > > ### Author Response · Authors · 2023-08-22
> > > > **Response to additional comments part 3**
> > > >
> > > > `L261: language mistakes (on the tasks, the authors of [80])`
> > > > Response: Could you please re-phrase your remark?
> > > >
> > > > `L279: What is the soundex algorithm?`
> > > > Response: Soundex is an old algorithm for indexing names based on their English pronunciation. The idea is to map homophones to the same code (consisting of a letter and a three-digit code) so that names that sound the same are mapped to the same code, even if they differ slightly in spelling.
> > > >
> > > > `L786: "In neither of the considered training scenarios (HL, HB, TF), a single model is statistically better than others.” Why isn’t this discussed in the main paper?`
> > > > Response: In a way, it is. Due to space constraints, we present a single benchmark result (Fig. 1, all models and all languages, fine-tuning scenario) as it is the most interesting for readers. Looking at the columns in Fig. 1, we can see that the differences between models are not big for a particular language, but the differences between languages are significant. More benchmark results for other scenarios (HB, HL) and other data slicing are presented in the supplement, along with the results of statistical tests. This type of benchmark is too complex to report a single finding; as we state in L.849, it really depends on multiple factors. The domain, the availability of similar languages, the mode of deployment, and the budget for additional fine-tuning all these factors influence the proper choice of a model and a deployment strategy. Our over-arching ambition is to introduce this dataset & benchmark as a meta-benchmark (for the lack of a better term), i.e., the helpful tool where other researchers can use our language and dataset typology to select appropriate datasets for training models for a particular use case.

---

### Official Review · Reviewer_98Yo · 2023-07-21
**Good multilingual Corpus for Sentiment Dataset but evaluation and limitations hinders the paper**

**Rating:** 5
**Confidence:** 4

**Strengths:**

1 - The paper presents a collection of a large multilingual dataset for sentiment analysis classification models
2 - The paper has a strong and convincing motivation.
3 - The paper addresses the limitations and ethical limitations.

**Additional Feedback:**

There is no additional feedback.

**Clarity:**

No, he paper is not well written. There are many sentences that state facts and they don't explain or justify why this is chosen or how, Also there are no examples provided in many places. For example in page 4, the quality criteria does not present examples of unchosen datasets. What exact issues do they have so they can be removed or not chosen?
- There is no 'conclusion' section. Is it the 'Discussion' Section?


**Correctness:**

The evaluation method is not clear enough for a reviewer to judge. For example, in section 3.2 (second paragraph). Did the authors only prepare a separate internal dataset with manual annotations in Polish and English only? If so, what about the rest of the languages? The introduction states why the task is challenging, but if the evaluation only follows the independent dataset, then the evaluation is inappropriate.

**Documentation:**

Section 2 (Linguistic typology): The first paragraph states many facts but without citations. If the authors state those facts from their knowledge, then some of the hypothesized facts could be challenged.  For example, see the sentences: "As of today, linguists define ...... . The largest families ...... . Where is the citation? This is just one example of many.

**Ethics:**

The authors have mentioned them all.

**Limitations:**

Yes, the authors have discussed the limitations of their work in 3 fold:
1 - The collection of the dataset is highly biased toward English.
2 - There exists a significant variance in sample quality across all datasets and all languages.
3 - The authors tried enforcing strict quality criteria and rejected many unqualified datasets, but they were unable to unify the annotation guidelines. Understandably, that is a challenging task.

**Opportunities For Improvement:**

1 - The paper's writing is hard to follow.
2 - Many stated facts and statements lack citation.
3 - A Comparison of similar works and other datasets to theirs is missing.
4 - The paper compiles available datasets into one. Therefore the originality of the dataset is not present.

**Relation To Prior Work:**

No, relation to prior work is not clearly discussed. There is no comparison between their dataset and available datasets. It is expected that the paper at least would present a table that compares their work to previous work in terms of size, diversity, and other aspects.

**Summary And Contributions:**

This paper presents a large multilingual corpus of datasets for training sentiment models. The dataset is a collection of available datasets. Further, the paper presents a multi-faceted sentiment classification benchmark to summarize several experiments of base models. However, the issues in this paper are not minimal and need to be addressed.

The paper's contributions are the following: 1 - The large multilingual collected sentiment classification datasets. It contains 79 selected datasets in 27 languages. 2 - The work presents a benchmark containing detailed run statistics of several experiments representing different training and testing scenarios. 3 - The dataset is publicly available and easily accessible.

---

> ### Author Response · Authors · 2023-08-09
> **Responding to Reviewer Feedback: Enhancing our Paper with Improved Dataset Protocol**
>
> We would like to express our gratitude for your work in reviewing our paper. Your review points to a few points that require improvement or clarification. Your remarks closely follow the suggestions of another reviewer regarding the protocol to collect, collate, and merge various sentiment datasets. We are committed to addressing these issues and refining our paper to meet the high standards of NeurIPS. We have started revising the paper to address most of your concerns. We will submit the paper's revised version, along with the supplementary material and detailed comments, on this forum by August 21. We want to give the reviewers sufficient time to evaluate the revision and our responses. Please accept our sincere thanks for your time and effort, and rest assured, your recommendations will be duly incorporated in our revision.

---

> ### Author Response · Authors · 2023-08-22
> **Response to review Good multilingual Corpus for Sentiment Dataset but evaluation and limitations hinders the paper part 1**
>
> Once again, thank you for a constructive review.
>
> We will link our responses to each of your comments for easier readability.
>
> `1 - The paper's writing is hard to follow.`
>
> Response: We appreciate your comment and we read carefully the paper several times, improving many parts. Please read other’s reviewer comments and our responses to check what has been updated. We hope it is a better flow of reading now.
>
>
>
> `2 - Many stated facts and statements lack citation.
>  3 - A Comparison of similar works and other datasets to theirs is missing.`
>
> Response: Thank you for pointing this out, we added more citations to Section 2 "Linguistic typology" and extended Section 5 with additional section 5.3.
>
> `4 - The paper compiles available datasets into one. Therefore the originality of the dataset is not present.`
>
> Response: By combining datasets into one collection, we remove the originality of constituent datasets. However, this issue is partially addressed as we augment our submission with a detailed list of all datasets we analyzed. The list of datasets contains links to original datasets and/or papers presenting them.
>
> Suppose the reviewer's primary concern is that our work does not present an original contribution because we are compiling existing datasets. In that case, we kindly disagree with such a perspective. The task of reviewing and analyzing hundreds of datasets was very labor-intensive. The result is synergetic; the compiled collection of datasets with linguistic typology and dataset features is far more helpful than any datasets alone. The dataset can be used to produce individualized benchmarks (for a language family, for a task, for data modality), the dataset is published using a very user-friendly API, and the construction of the dataset allows for easy extension of the dataset - users may create issues in the GitHub repository providing required meta-data. New datasets can be easily added to the collection, and we also provide an idea of how they could be added here https://github.com/Brand24-AI/mms_benchmark/issues/4. Moreover, we will review and update the corpus according to issues added in the GitHub repository to make this process a continual effort to help the NLP and sentiment community use an up-to-date version of the corpus.
>
> We also wanted to highlight the problem of creating one single, objective benchmark using such many languages, domains, family languages, etc. Our benchmarking part of the paper is an idea of how it could be benchmarked. However, we wanted to enable others to use sentiment datasets for their own purposes as easily as it could be possible. Now, you can slice and dice different datasets with a couple of lines of Python code and you will not need to spend days or even months looking for datasets and then clean them and unify them for your experiments.
>
> Correctness:
> `The evaluation method is not clear enough for a reviewer to judge. For example, in section 3.2 (second paragraph). Did the authors only prepare a separate internal dataset with manual annotations in Polish and English only? If so, what about the rest of the languages? The introduction states why the task is challenging, but if the evaluation only follows the independent dataset, then the evaluation is inappropriate.`
> Response:  There is no evaluation “per se” presented in the paper. The idea of the “internal” dataset was not explained clearly in the original submission and has been improved in the revision - Section 3.2. Basically, the “internal” dataset serves as a baseline which allows us to compare various combinations of models’ hyperparameters. It was compiled for two languages only because the annotators were fluent in these languages, and the dataset has been used internally in Brand24. It should not be treated as a strict validation; rather one should think of it as a manually curated, high-quality dataset with two very different languages on which multilingual models can be reasonably compared. Of course, creating such a baseline for all 27 languages is impossible given our resources. We needed a high-quality dataset with full control over the annotation protocol, aligned with the domains operated by Brand24, and useful for comparing different model configurations.
> The revised version of the paper contains an extended section on experiments, where we present more data splits and pivots of languages, models, and sentiment classes. This extended section can be regarded as a more proper validation of the usefulness of our dataset and our benchmark.

---

> > ### Author Response · Authors · 2023-08-22
> > **Good multilingual Corpus for Sentiment Dataset but evaluation and limitations hinders the paper part 2**
> >
> > Clarity:
> > `No, he paper is not well written. There are many sentences that state facts and they don't explain or justify why this is chosen or how, Also there are no examples provided in many places. For example in page 4, the quality criteria does not present examples of unchosen datasets. What exact issues do they have so they can be removed or not chosen?`
> >
> > Response: This issue has been raised by other Reviewers as well and prompted us to supplement the submission with a full detailed presentation of datasets considered for inclusion in our corpus. For each dataset, we report the size of the dataset, the distribution of sentiment labels, the link to a paper/repository with the source data, and any meta-data we could extract from the dataset. Each dataset is characterized by the reason for rejection if the dataset has been indeed rejected. We hope that this addition fully addresses your concern.
> >
> > `There is no 'conclusion' section. Is it the 'Discussion' Section?`
> > Response: This was a deliberate choice when preparing the submission. There are no clear conclusions stemming from our work, we present the research community with a unique multilingual dataset, an API for extremely easy data selection/filtering, and a benchmark of current SOTA models for multilingual sentiment classification. Some conclusions are indeed reported in the “Discussion”, and some are presented in Appendix B “Experiments”, where we present results of statistical comparison of models on different languages and tasks.
> > Relation To Prior Work:
> >
> > `No, relation to prior work is not clearly discussed. There is no comparison between their dataset and available datasets. It is expected that the paper at least would present a table that compares their work to previous work in terms of size, diversity, and other aspects.`
> > Response: This is an embarrassing omission on our part, we have included the related work on multilingual language models and multilingual sentiment classification, but we have forgotten about multilingual sentiment datasets. We have added Section 5.3 with the description of several previous multilingual datasets with sentiment labels.
> > Documentation:
> >
> > `Section 2 (Linguistic typology): The first paragraph states many facts but without citations. If the authors state those facts from their knowledge, then some of the hypothesized facts could be challenged. For example, see the sentences: "As of today, linguists define ...... . The largest families ...... . Where is the citation? This is just one example of many.`
> >
> > Response: The statistics on human languages were taken from “The Ethnologue” and “The Atlas of Languages”. Information about the close relationship between languages and human DNA trees has been published in Science. Information about contacts between languages has been found in a seminal book on linguistics by Thomason and Kaufman. All relevant citations have been inserted into the text.

---

### Official Review · Reviewer_HqfF · 2023-07-22
**An interesting benchmark but lacking detailed analysis**

**Rating:** 5
**Confidence:** 4
**Correctness:** Yes
**Clarity:** Yes

**Strengths:**

* The investigated problem is interesting where multilingual sentiment classification datasets can be a good lens to analyze the cultural difference in expressing sentiments across different countries.
* Multiple models are taken to evaluate against this newly collected benchmark.

**Additional Feedback:**

Line 30: multilingual => Multilingual

**Documentation:**

Yes

**Limitations:**

The contributions of this paper are quite weak. The authors mainly present basic statistics of the collected datasets and the main results of some selected models on this benchmark. The readers cannot obtain any useful insights from this paper.

Alternatively, a more detailed analysis of the constructed benchmark should be conducted. For example, why the filtered dataset has higher quality? What are the typical expressions in terms of different sentiment polarities in each language? Similarly, more analysis around those adopted models should be conducted.

**Opportunities For Improvement:**

See below

**Relation To Prior Work:**

Discussion on existing multilingual or monolingual sentiment classification datasets is insufficient.

**Summary And Contributions:**

This paper introduces a multilingual sentiment classification benchmark that collects various (79 in total) sentiment classification datasets across 27 languages. Then multiple multilingual models are evaluated on this dataset. Empirical results show that performance on low-resource languages varies a lot compared to high-resource languages.

---

> ### Author Response · Authors · 2023-08-09
> **Addressing Reviewer Feedback: Enhancing our Paper to Meet Expectations**
>
> We are grateful for your time and effort in reviewing our work. Upon receiving your evaluation, we must admit that we are somewhat disheartened to learn that our paper did not meet your expectations. It is a regrettable situation, as we have devoted significant time and effort to this project. Nonetheless, we understand and respect your professional perspective and are eager to take this as an opportunity for learning and improvement. We have obtained quite detailed reviews with multiple questions and suggestions, and we are now working on a significant rewrite of the paper. Hopefully, the amended version of the paper will address some of your concerns and criticisms. We cordially invite you to follow the discussion with the remaining reviewers, as these comments might dispel your doubts regarding our paper.

---

> ### Comment · Area_Chair_gMHE · 2023-08-15
>
> Reviewer HqfF, can you please provide more constructive feedback about how the paper can be improved? Why do you say that "The readers cannot obtain any useful insights from this paper"?
>
> (Opportunities for Improvement is currently blank, and should be actionable steps that the authors can take in order to improve upon the identified Limitations)

---

> ### Comment · Reviewer_HqfF · 2023-08-25
> **Comments after checking the revisions**
>
> Upon reviewing the revisions, I discovered that the authors have incorporated more analysis, both for the newly compiled dataset and benchmarking analysis. This addresses some of my concerns mentioned in the 'Limitations' section. Consequently, I raised my score to 5.
>
> I understand that the authors have chosen to place greater emphasis on the dataset component, which has resulted in a less extensive analysis of the performance of various models within the paper. However, I would still encourage the authors to integrate more analysis into the main body of the paper, which can better show the usefulness of this dataset, e.g., how challenging it is and what limitations of models it can reveal.

---

### Official Review · Reviewer_5m3s · 2023-07-25
**Significant amount of work, needs more detail in the paper though.**

**Rating:** 7
**Confidence:** 5
**Clarity:** The paper is pretty clear in terms of…

**Strengths:**

- Multilingual SA benchmark, multifaceted too.
- The benchmarking/evaluation is sufficiently exhaustive.
- Quality checks seem stringent, leading to only 79 datasets being considered out of 350.

Overall, significant contributions from the paper in terms of dataset and benchmarking.

**Additional Feedback:**

None

**Correctness:**

The claims made in the submission are correct but the paper draft needs to be improved given the suggestions above.

**Documentation:**

The benchmarking done lists sufficient metadata for each dataset. There is sufficient data for reproducing the experiments.

**Ethics:**

No ethical concerns.

**Limitations:**

Yes, the authors have addressed the limitations for this work in sufficient detail.

**Opportunities For Improvement:**

- The complete results of the benchmarking are not described in the paper-- Some are in the appendix, and some are on the HuggingFace page. The paper can be reorganized with more results as a part of the main draft, or the appendix. I suggest you shift the linguistic typology analysis to the appendix and describe the complete benchmarking results in the main draft.
- Discussion falls short on error analysis which the authors should show. Given you are utilizing SoTA encoding approaches for benchmarking, it would be interesting to look at what sentiment classes are not being captured sufficiently and from which languages. You have not highlighted any language or any particular datasets as challenging, which should have been done.


**Relation To Prior Work:**

Yes, the prior work has been described well, and all seems to be cited too.

**Summary And Contributions:**

This paper introduces a multilingual dataset for the sentiment analysis task in more than 27 languages (6 language families). The data comprises 79 existing datasets, which have been sourced from GitHub, HuggingFace, and so on. The paper additionally describes the work on benchmarking these datasets using a multilingual sentiment analysis model using fine-tuning with a linear classification head; and with a BiLSTM head, for the sentiment classification task. The results described in the paper show the performance of the multilingual model after fine-tuning them over the complete dataset. The results are convincing for most models, but some models show the performance capped at 41-45, slightly above the random baseline for a three-class classification (pt language). The multi-faceted benchmark description is present in the paper, but I do not see any evaluation or benchmarking based on this criteria (assuming this is in the works). From the appendix, it seems like the whole test set performance is quite good for the top 3 models. Further, the paper also contributes a Polish dataset (with significant agreement from annotators in terms of Cohen's kappa and Krippendorf's alpha), which they choose to keep internal and show the performance of their multilingual model on this data too. Overall, the contributions are significant for the paper to be considered but there are some flaws in the draft, which I will try to outline below.

---

> ### Author Response · Authors · 2023-08-09
> **Responding to Reviewer Feedback: Enhancing our Research Paper with Detailed Error Analysis and Revised Structure**
>
> We appreciate the time and effort you have invested in reviewing our work. Your critique and suggestions are insightful, and for that, we are grateful. We are committed to refining our work along with your remarks and comments from other reviewers, and we will start working on the necessary revisions. We plan to present an amended version of our paper by August 21st so that you have sufficient time to judge the completeness of the revision. Thank you once again for your constructive feedback.
>
> We will include a much more detailed error analysis and description of the benchmarking experiments. We have also considered your suggestion to re-organize the paper and move the linguistic typology to an appendix, focusing instead on the benchmark. This was a subject of lively discussion among the authors. The track is aptly named "Datasets & Benchmarks", and it seems that the Authors and Reviewers have a slight joint bias toward benchmarks. Our paper introduces both: we consider the compilation of the multilingual and multimodal sentiment dataset to be an equal contribution to the computational and experimental effort of benchmarking models. We would like the paper to reflect this dual contribution - the typology can be helpful for other researchers, allowing them to easily create sub-datasets using various dimensions of our dataset and creating more focused domain benchmarks for sentiment classification models. As of this writing, we feel more inclined to keep the current macro-structure of the paper divided roughly 50-50 between the description of the dataset and the description of the benchmark. However, we do not exclude that at the stage of preparation of the article's final version, this structure will not change.

---

> ### Author Response · Authors · 2023-08-22
> **Response to Significant amount of work, needs more detail in the paper though**
>
> Once again, thank you for a constructive and warm review. We are glad you like our contribution to the NLP community and fellow sentiment researchers. We hope it will make conducting sentiment analysis experiments and new benchmarks easier.
>
> We will link our responses to each of your comments for easier readability.
>
> Opportunities For Improvement:
> `The complete results of the benchmarking are not described in the paper-- Some are in the appendix, and some are on the HuggingFace page. The paper can be reorganized with more results as a part of the main draft, or the appendix. I suggest you shift the linguistic typology analysis to the appendix and describe the complete benchmarking results in the main draft.`
>
> Response: This point has been discussed by the Authors at length. We have decided to leave the original structure of the paper and provide more experimental results in the supplementary materials section. Our rationale is as follows: our work has two equally important contributions. The benchmark is obvious: it is a valuable guide for other researchers on which model to select given the language (or languages) and the domain (+ which deployment scenario might work best). However, equally important is the dataset itself. It gives the researchers a unique possibility to create data splits along many linguistic and functional dimensions quickly. Researchers may use this resource to build domain benchmarks (e.g., the ranking of Twitter sentiment classifiers for Romance languages). The work put into searching, evaluating, selecting, filtering, and merging datasets was significant, and we want this to be reflected in the structure of the paper. Also, from a purely practical point of view, the interface to quickly select, filter and download datasets for model training is even more helpful than the benchmark itself.
>
> `Discussion falls short on error analysis which the authors should show. Given you are utilizing SoTA encoding approaches for benchmarking, it would be interesting to look at what sentiment classes are not being captured sufficiently and from which languages. You have not highlighted any language or any particular datasets as challenging, which should have been done.`
>
> Response: The Supplementary Materials section has been extended with more examples of experiments conducted on the benchmark dataset. In particular, we present detailed classification results divided along language and domain sentiment, identifying the most challenging cases where multilingual sentiment models fail. Our dataset allows to create numerous data splits, and each can be used as a benchmark dataset for a specific scenario. For instance, we can quickly select Slavic languages with data extracted from news sites to improve the sentiment classifier for online comments. If there is one general finding resulting from our work, then it would be “no single multilingual model performs best across languages, tasks, and domains”.
> In addition, we also added the same results in the form of spreadsheets to the supplementary materials. It could be easier to analyze them or create your own pivot tables.

---

### Author Response · Authors · 2023-08-22
**General response to all reviewers**

Thank you (all reviewers) for the valuable feedback. Firstly, we have a couple of updates for all of you.

Why do we not focus only on the benchmarking part of our paper and leave the initial structure as it is? We have discussed this point at length. We have finally decided to leave the original structure of the paper and provide more experimental results in the supplementary materials section. Our rationale is as follows: our work has two equally important contributions. The benchmark is obvious: it is a valuable guide for other researchers on which model to select given the language (or languages) and the domain (+ which deployment scenario might work best). However, equally important is the dataset itself. It gives the researchers a unique possibility to create data splits along many linguistic and functional dimensions quickly. Researchers may use this resource to build domain benchmarks (e.g., the ranking of Twitter sentiment classifiers for Romance languages). The work put into searching, evaluating, selecting, filtering, and merging of datasets was significant, and we want this to be reflected in the structure of the paper. Also, from a purely practical point of view, the interface to quickly select, filter and download datasets for model training is even more useful than the benchmark itself, especially if you can do it all in a couple of lines of Python code (see Appendix Section C - Listings for examples). It can accelerate experimentation and create new benchmarks heavily.

In addition, we prepared a spreadsheet with datasets that we collected and filtered for the final corpus of datasets. We checked even more sources of potential datasets that are in the spreadsheet, but many didn’t contain the option to download, or there was no information about related papers or any other description of the dataset. We want to use this collection to add and test if new datasets are worth adding to the corpus. Please check supplementary data - “MMS - datasets filtering.xlsx”.

Moreover, constructing our corpus allows for easy dataset extension - users may create issues in the GitHub repository providing the required metadata. New datasets (for example, Greek language datasets) can be easily added to the collection, and we also provide an idea of how they could be added here [Dataset suggestion]: Homeric Text: The 1st Book of Iliad · Issue #4 · Brand24-AI/mms_benchmark (github.com). We will review and update the corpus according to issues added in the GitHub repository to make this process a continual effort to help the NLP and sentiment community use an up-to-date version of the corpus.

We have already added the first Greek dataset to the MMS Corpus. It could be easily used with this code snippet

```python
import datasets
mms_dataset = datasets.load_dataset("Brand24/mms", download_mode="force_redownload") # it case you have already downloaded the corpus, we must refresh HF cache
greek = mms_dataset.filter(lambda row: row['language'] == 'el')
```

We hope our labor-intensive work will accelerate the research of fellow sentiment researchers. It could also influence the reproducibility of research because it will be a fixed version of datasets with a single place to download and load them. Less duplicated work and more reproducible experiments.

Finally, we also added an extended error analysis of the benchmark results to the appendix D, with all results in an editable form available here https://drive.google.com/drive/folders/1JKBlslTXXiTVu9cHhQL7Fo4GdhUmdInx?usp=sharing and zipped in the supplementary materials.

---

### Decision · Program_Chairs · 2023-09-22

**Decision:**

Accept (Poster)

**Comment:**

The submission presents interesting and extensive work. The authors improved many aspects of their paper based on reviewer comments and have put in a lot of additional work in making it clearer and easier to understand. The fact that the proposed datasets are already up and ready for download via `datasets` makes them user-friendly and a clear contribution to the community.